# PRMT5 is essential for B cell development and germinal center dynamics

Ludivine C. Litzler[1,2], Astrid Zahn[1], Alexandre P. Meli[3], Steven Hébert [4], Anne-Marie Patenaude[1,12], Stephen P. Methot [1,13], Adrien Sprumont[1], Thérence Bois[1], Daisuke Kitamura [5], Santiago Costantino [6,7], Irah L. King[3], Claudia L. Kleinman[4,8], Stéphane Richard[4,9] & Javier M. Di Noia [1,2,10,11]

Mechanisms regulating B cell development, activation, education in the germinal center (GC) and differentiation, underpin the humoral immune response. Protein arginine methyltransferase 5 (Prmt5), which catalyzes most symmetric dimethyl arginine protein modifications, is overexpressed in B cell lymphomas but its function in normal B cells is poorly defined. Here we show that Prmt5 is necessary for antibody responses and has essential but distinct functions in all proliferative B cell stages in mice. Prmt5 is necessary for B cell development by preventing p53-dependent and p53-independent blocks in Pro-B and Pre-B cells, respectively. By contrast, Prmt5 protects, via p53-independent pathways, mature B cells from apoptosis during activation, promotes GC expansion, and counters plasma cell differentiation. Phenotypic and RNA-seq data indicate that Prmt5 regulates GC light zone B cell fate by regulating transcriptional programs, achieved in part by ensuring RNA splicing fidelity. Our results establish Prmt5 as an essential regulator of B cell biology.

[1] Institut de Recherches Cliniques de Montréal, 110 av. des Pins Ouest, Montréal, QC H2W 1R7, Canada. [2] Department of Biochemistry and molecular medicine, 2900 boul. Édouard-Montpetit, bureau D-360, Montréal, QC H3T 1J4, Canada. [3] Meakins-Christie Laboratories, Department of Microbiology and Immunology, McGill University Health Centre, 1001 boul. Decarie, Block E (EM2.2232), Montreal, QC H4A 3J1, Canada. [4] Segal Cancer Center, Lady Davis Institute for Medical Research, Montréal, Québec H3T 1E2, Canada. [5] Research Institute for Biomedical Sciences, Tokyo University of Science, Noda, Chiba 278-0022, Japan. [6] Research Center of the Hospital Maisonneuve-Rosemont, 5415 Boulevard de l'Assomption, Montréal H1T 2M4 QC, Canada. [7] Department of Ophthalmology, Université de Montréal, C.P. 6128, succ. Centre-ville, Montréal, QC H3C 3J7, Canada. [8] Department of Human Genetics, Faculty of Medicine, McGill University, 1205 Dr. Penfield Ave, Montréal, Quebec H3A 1B1, Canada. [9] Departments of Oncology and Medicine, McGill University, 5100 Maisonneuve Blvd West, Suite 720, Montreal, QC H4A3T2, Canada. [10] Department of Medicine, Division of Experimental Medicine, McGill University, 1001 boul Decarie, Montreal, QC H4A 3J1, Canada. [11] Department of Medicine, Université de Montréal, C.P. 6128, succ. Centre-ville, Montréal, QC H3C 3J7, Canada. [12] Present address: Genos, BioCentar Borongajska cesta 83H, 10000 Zagreb, Croatia. [13] Present address: Friedrich Miescher Institute for Biomedical Research, R-1066.2.58. P.O. Box 3775, Maulbeerstrasse 66, 4002 Basel, Switzerland. Correspondence and requests for materials should be addressed to J.M.D.N. (email: javier.di.noia@ircm.qc.ca)

B lymphocytes transit through multiple cellular stages to acquire functional proficiency and produce high affinity antibodies. B cell development in the bone marrow (BM) alternates between quiescent and replicative stages, with checkpoints for the successful rearrangement of the immunoglobulin genes (*Ig*)[1]. Mature B cells in the periphery undergo further functional transformations after cognate antigen engagement: activation, proliferation, programmed *Ig* mutation coupled to antibody affinity-based selection in the germinal center (GC), and differentiation into memory or plasma cells[2].

The transition of mature B cells from quiescence to an activated state requires functional changes enabled by rapid transcriptional changes[3]. T-cell help stimulates migration of activated B cells into lymphoid follicles, where proliferation drives the GC reaction. The GC undergoes formation, expansion, and attrition over ~3 weeks after antigenic challenge[2]. Mature GCs are organized into two separate regions, the dark (DZ) and light (LZ) zones, which contain functionally distinct B cell subsets[2]. Centroblasts in the DZ are highly proliferative and undergo *Ig* somatic hypermutation initiated by activation-induced deaminase (AID). Centrocytes in the LZ proliferate less and compete for antigen and T cell help, which select those expressing high-affinity antibodies[4]. These functional changes during the GC reaction are regulated by master transcription factors including Bcl6 and Pax5 that define the GC fate, while the expression of Irf4 and Prdm1 defines plasma cell differentiation[5]. In contrast, transcriptional differences between centrocytes and centroblasts are subtle[6]. Nevertheless, additional transcriptionally defined GC B cell subsets suggest a more than binary GC dynamics[7,8].

Gene expression is regulated by post-translational modifications of chromatin components, including arginine methylation catalyzed by a family of protein arginine methyltransferases (PRMTs) that can also regulate pre-mRNA processing, protein synthesis, and signal transduction[9,10]. The relevance of arginine methylation in B cells was suggested by a pan-PRMT inhibitor, which reduced B cell proliferation ex vivo[11]. However, enzyme-specific analyses are necessary, as each PRMT modifies a non-overlapping set of substrates and mice lacking individual PRMTs display different phenotypes[9]. There are three types of PRMTs. Type I PRMTs transfer two methyl groups to the same nitrogen of the arginine guanidino group to produce asymmetric dimethyl-arginine (DMA), type II produce symmetric DMA (sDMA) by modifying two different nitrogen atoms, and type III transfer a single methyl group[9]. Recent work on two PRMTs indicates that each has unique functions in B cells. The type I methyltransferase PRMT1 promotes Pre-B cell differentiation and is necessary for GC formation and antibody responses[12–15]. The type III methyltransferase PRMT7 limits GC formation[16]. Little is known about the role of the type II enzymes PRMT5 and PRMT9 in normal B cells, but Prmt5 and sDMA levels are increased in activated mouse B cells[17], suggesting a physiological function.

PRMT5 has garnered attention because it is overexpressed in GC-experienced and mantle cell human B cell lymphomas, correlating with poor prognosis[18,19]. Accordingly, PRMT5 promotes disease progression in mouse models of oncogene-driven leukemia[20] and its depletion reduces proliferation of B cell lymphoma cells[18,19,21]. PRMT5 inhibition is emerging as a potential therapy against lymphoma[22,23] calling for understanding the relevance and functions of this enzyme in normal B cells.

PRMT5 is responsible for most cellular sDMA and has multiple substrates, which allow PRMT5 to regulate major aspects of cell physiology[24]. PRMT5 acts mainly as a transcriptional corepressor by methylating histones but can also regulate the function of transcription factors, notably p53[19,24]. PRMT5 also methylates splicing factors to modulate pre-mRNA processing[19,25,26], as well as cytoplasmic proteins to regulate signaling[27]. Additionally, PRMT5 can regulate homologous recombination-mediated DNA repair[28].

Here we show that Prmt5 is critical for all major proliferative B cell stages during development in the BM and periphery, as well as for antibody responses. Prmt5 regulates transcription and splicing fidelity in B cells, thereby preventing an apoptotic p53 response that otherwise hampers the development of Pro-B cells. Moreover, Prmt5 has critical p53-independent functions in most other B cell stages, underpinning Pre-B cell differentiation and preventing apoptosis during B cell activation and promoting GC expansion, with the latter function mediated, at least in part, by negatively regulating plasma cell differentiation. Our findings uncover Prmt5 as an important determinant of GC dynamics and will enable the comparison of the functions of Prmt5, and the effects of its inhibition, in normal and cancerous B cells.

## Results

**Distinct regulation of Prmt5 in proliferating B cells.** To infer the time points in which Prmt5 might be functionally relevant, we analyzed *Prmt5* expression in B cell subsets. Prmt5 mRNA peaked during B cell development at the Pro-B and early Pre-B cell stages (Fig. 1a). Most subsequent stages, including all mature B cell populations, expressed similar Prmt5 mRNA levels. By contrast, Prmt5 mRNA was upregulated in B cells activated ex vivo (Fig. 1a). Accordingly, Prmt5 protein and sDMA detected on multiple proteins were maximal 48 h after stimulating resting splenic B cells with lipopolysaccharide (LPS) and IL-4 (Fig. 1b). Consistent with the gene expression profile, conditional deletion of *Prmt5* in Pro-B cells completely blocked B cell development (see below). However, using the CD19-cre driver, which deletes inefficiently in the BM[29], allowed B cell development (Supplementary Fig. 1A). *Prmt5*$^{F/F}$ CD19-cre mice produced normal numbers of splenic B cell subsets (Fig. 1c, Supplementary Fig. 1B) that were depleted of Prmt5 (Fig. 1d). Prmt5 protein and sDMA levels were also reduced in *Prmt5*$^{F/F}$ CD19-cre splenic B cells stimulated ex vivo (Fig. 1e), with Prmt5 mRNA and protein levels correlating well (Fig. 1f). In contrast to activated B cells, GC B cells showed similar Prmt5 mRNA levels to follicular B cells (Fig. 1a) but immunohistochemistry (IHC) revealed high Prmt5 expression within splenic GCs in immunized mice (Fig. 1g). Publicly available IHC data confirmed higher Prmt5 expression in GC versus follicular B cells in human lymphoid tissues (Supplementary Fig. 1C). Thus, Prmt5 is dispensable for the homeostasis of splenic follicular and marginal zone B cells but upregulated in proliferating B cell stages, with distinct regulation in activated and GC B cells that suggest different functions in each of these stages.

**Prmt5 promotes survival of activated B cells in vivo.** To assess the relevance of Prmt5 in stimulated B cells, we immunized mice with NP-CGG in alum. GC B cell numbers were reduced by >2-fold in *Prmt5*$^{F/F}$ CD19-cre compared to control mice at day 14 post-immunization (Fig. 2a). Despite Prmt5 being efficiently deleted in follicular B cells in *Prmt5*$^{F/F}$ CD19-cre mice, the vast majority of GC B cells found after immunization were consistently Prmt5$^+$ (Fig. 2b, Supplementary Fig. 1D). Quantification of the immunofluorescence (IF) signal showed significantly reduced Prmt5 levels in follicular B cells from *Prmt5*$^{F/F}$ CD19-cre mice compared to controls, while GC B cells displayed similar Prmt5 levels in both mice, resulting in a significantly higher follicular/GC Prmt5 signal ratio (Fig. 2b). We conclude that the minority of B cells that had failed to excise *Prmt5* outcompeted Prmt5-null B cells after activation in these mice, strongly suggesting a B cell intrinsic role for Prmt5 in GC formation.

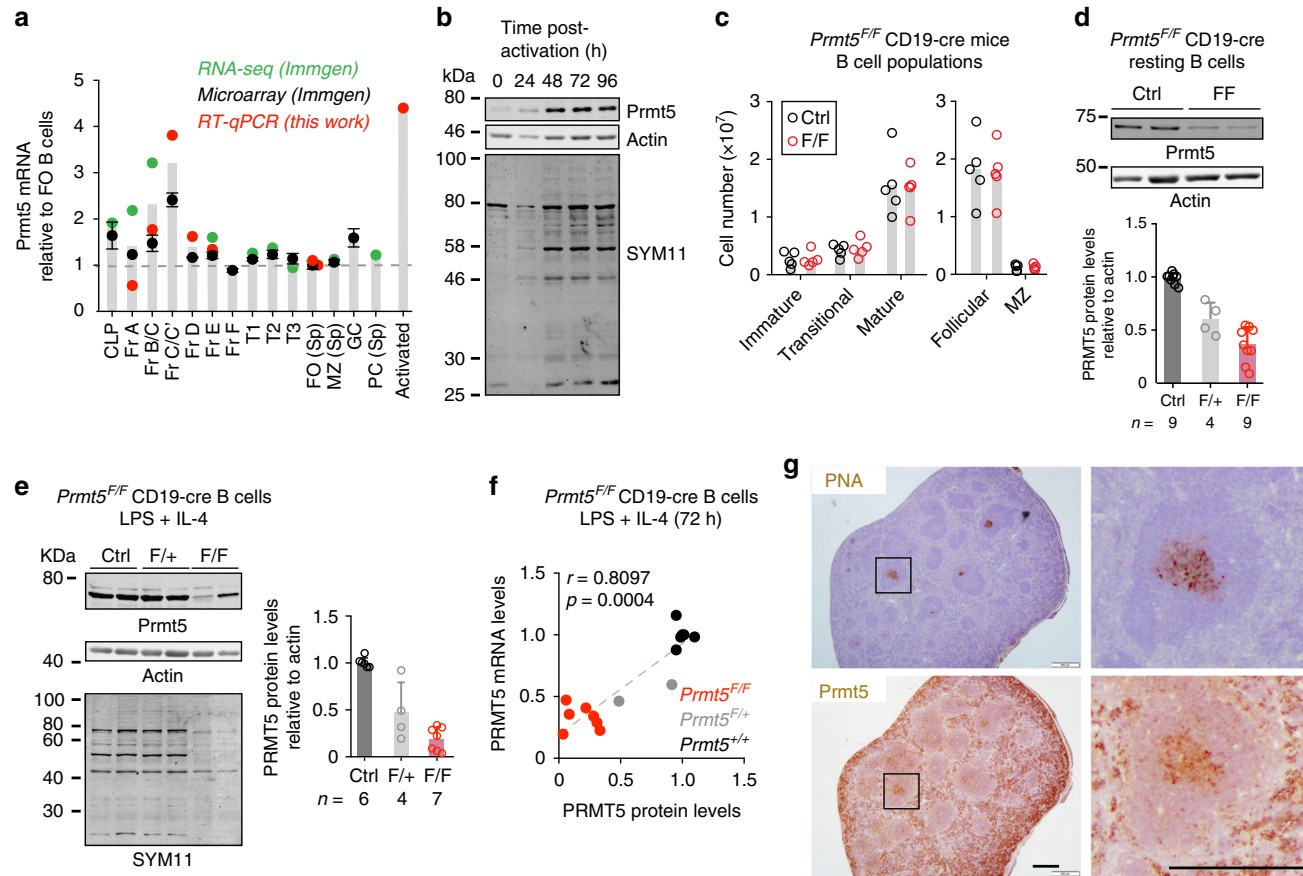

**Fig. 1** Regulated Prmt5 expression in B cells. **a** Prmt5 transcript levels in B cell stages from the indicated datasets, each normalized to follicular (Fo) B cells. RT-qPCR data were normalized to Actin mRNA and obtained from a pool of two mice sorted for Hardy's BM fractions (FrA to E), or from splenic B cells from two mice stimulated ex vivo with LPS and IL-4 for 48 h (Activated). CLP common lymphoid progenitor, T1–T3 transitional B cells, MZ marginal zone B cells, PC plasma cells, Sp spleen. **b** Prmt5 expression kinetics and sDMA-modified proteins by WB in extracts of WT splenic B cells stimulated with LPS (5 μg/mL) and IL-4 (5 ng/mL) probed with anti-PRMT5, -Actin (as loading control), and -sDMA (SYM11) antibodies. **c** Absolute number of B cell subpopulations in the spleen of CD19-cre (Ctrl) and Prmt5^{F/F} CD19-cre (F/F) mice. Individual mice (dots) and mean (bars) values are plotted. Gatings in Supplementary Fig. 1B. **d** Representative WB of resting splenic B cell extracts from CD19-cre (Ctrl) and Prmt5^{F/F} CD19-cre (F/F) mice. Means + s.d. levels of Prmt5 normalized to Actin quantified from n mice by WB are plotted relative to the Ctrl. **e** Representative Prmt5, Actin, and sDMA (SYM11) WB and quantitation as in **d**, in extracts of splenic B cells activated with LPS (5 μg/mL) and IL-4 (5 ng/mL) for 72 h. **f** Prmt5 mRNA (by RT-qPCR) as a function of Prmt5 protein (by WB) at 72 h post-stimulation for individual mice. Spearman's test correlation coefficient (r) and p-value (p) are indicated. **g** Representative immunohistochemical staining for PNA as GC marker and Prmt5 on consecutive mouse spleen sections at day 14 post-immunization. Scale bars, 500 μm

We were intrigued by the reduction in total lymphocyte count due to B cell-loss in immunized Prmt5^{F/F} CD19-cre mice, which was not observed in unimmunized mice (Fig. 2c). Alum causes some polyclonal B cell activation[30], as shown by the upregulation of the activation marker GL7 after injecting alum alone in our mice (Fig. 2d). To test the importance of Prmt5 upon B cell activation in vivo, we infected mice with *Heligmosomoides polygyrus*, a parasitic enteric nematode that induces polyclonal B cell activation[31]. At day 14 post-infection, total B cell numbers in mesenteric lymph node (MLN) and spleen were significantly reduced in Prmt5^{F/F} CD19-cre but not in control mice, whilst T cells were not affected (Fig. 2e). This reduction correlated with a significant increase in apoptosis in non-GC B cells in Prmt5^{F/F} CD19-cre mice (Fig. 2f). In contrast, neither T cells nor the mostly Prmt5+ GC B cells in these mice, showed any increase in apoptosis (Fig. 2f). Interestingly, most caspase activation was found in GL7− splenic B cells in these mice (Fig. 2f), suggesting that apoptosis occurred soon after activation and Prmt5-deficient B cells did not survive to become GL7+. We conclude that Prmt5 protects early activated B cells from apoptosis.

**Prmt5 prevents B cell apoptosis at activation.** To pinpoint the time at which Prmt5 was required during activation, we stimulated splenic B cells from Prmt5^{F/F} CD19-cre mice ex vivo with LPS and IL-4. Cell enumeration and cell division tracking dye showed reduced proliferation of Prmt5-deficient B cells (Fig. 3a). These Prmt5-null B cells had normal cell cycle profile (Fig. 3b) but significantly increased cell death, which was inversely correlated to Prmt5 levels (Fig. 3c, d). To test whether B cell survival depended on the catalytic activity of Prmt5, we treated wt B cells with the PRMT5-specific inhibitor EPZ015666 (EPZ)[23], which reduced sDMA in cell extracts (Supplementary Fig. 2A). Inhibition of Prmt5 24 h prior to activation increased cell death and reduced cell numbers after activation in an EPZ dose-dependent manner (Supplementary Fig. 2B). In contrast, adding EPZ simultaneously with LPS and IL-4 reduced proliferation but did not increase apoptosis compared to control (Supplementary Fig. 2C). This difference suggested that the anti-apoptotic function of Prmt5 was mediated by factors that were pre-methylated in resting B cells. To confirm this, we produced Prmt5^{F/F} Cγ1-cre mice, which express the Cre-recombinase

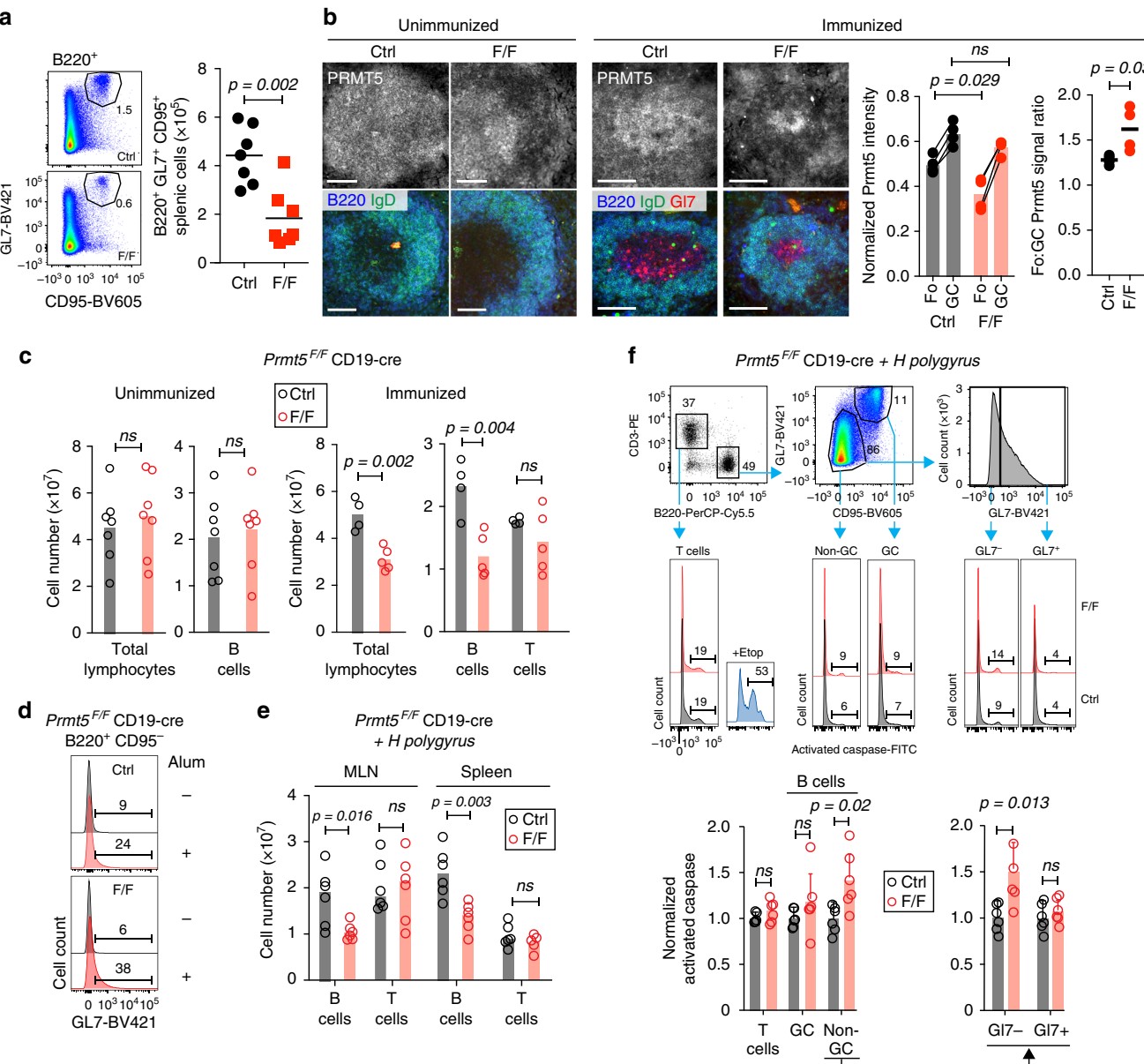

**Fig. 2** Prmt5 is required for the survival of activated B cells in vivo. All panels compare *Prmt5*^*F/F* CD19-cre (F/F) versus CD19-cre (Ctrl) mice. **a** Representative flow cytometry plots of splenic GC B cells (B220+, GL7high, CD95+) 14 days after NP-CGG immunization. Total number of GC B cell number for individual mice (symbols) and mean values (bars) from three experiments are plotted. **b** Representative immunofluorescence microscopy of mouse spleen sections stained for Prmt5, B220, GL7, and IgD from unimmunized (two per genotype from one experiment) or immunized with NP-CGG, day 14 (four per genotype from two experiments). Scale bars, 100 μm. The Prmt5 signal per follicular (Fo) and GC B cell was quantified in individual follicles from four immunized mice of each genotype. The lefthand plot shows mean normalized Prmt5 signal in Fo and GC B cells from 4–7 follicles per mouse (symbols), with lines joining mean values (bars) of the two B cell types in individual mice. The righthand plot shows the mean GC/Fo signal ratio per mouse. **c** Absolute number of splenic lymphocytes for individual mice unimmunized or 14 days post-immunization (symbols) with means (bars) from two experiments. **d** Representative histograms of GL7 expression in non-GC B cells (B220+ CD95−) from mice injected or not with Alum (day 14). One experiment, two mice per genotype per treatment. **e** Absolute number of B and T cells in mesenteric lymph node (MLN) and spleen of individual mice (symbols) 14 days post-infection with *H. polygyrus*, with medians (bars), from two experiments. **f** Sequential gating to analyze T, GC, and non-GC B cells, as well as the GL7− and GL7+ fractions of the latter (top) and representative histograms of pan-caspase staining in MLN of the mice analyzed in **e**. Gates were set using cells treated with etoposide (3 μM). Means + s.d. proportion of pan-caspase+ cells from six mice from two experiments are plotted, normalized to the control's average for each subset. *p*-Values are by Mann–Whitney test (**b**) or an unpaired, two-tailed Student's *t* test (**c**, **e**, **f**); ns, not significant

only after B cell activation[32]. We reasoned that the presence of the sDMA at the time of activation in *Prmt5*^*F/F* Cγ1-cre B cells might prevent the apoptosis seen in the sDMA-depleted *Prmt5*^*F/F* CD19-cre B cells. Indeed, stimulated *Prmt5*^*F/F* Cγ1-cre B cells

showed efficient Prmt5 depletion without any increase in apoptosis compared to control cells (Fig. 3e). We conclude that Prmt5 functions to protect B cells from apoptosis at the time of activation.

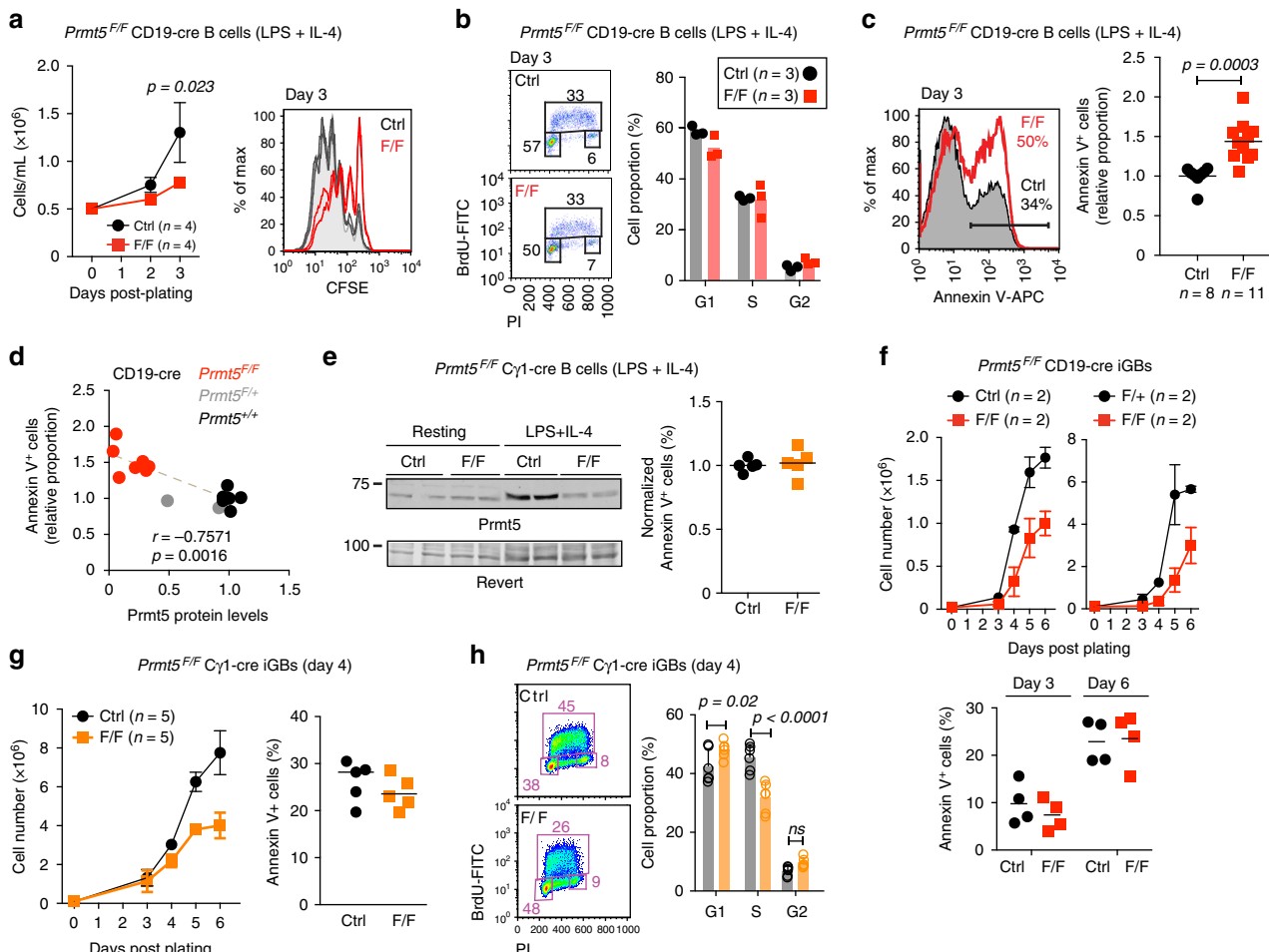

**Fig. 3** Prmt5 protects B cells from apoptosis and promotes proliferation. **a–d** One million naïve splenic B cells from CD19-cre (Ctrl) and *Prmt5$^{F/F}$* CD19-cre (F/F) mice were plated with LPS (5 μg/mL) + IL-4 (5 ng/mL). **a** Cell expansion over time monitored by cell counting (left) and number of cell divisions by day 3 monitored by CFSE stain dilution (right) from two experiments, four mice per genotype. **b** Cell cycle profile of cells pulsed with BrdU at day 3 before staining with anti-BrdU and propidium iodide (PI). Data for three mice (symbols) per genotype and means (bars) from one experiment are plotted. **c** Representative Annexin-V$^+$ staining histograms of B cells 3 days post-plating. The proportion of Annexin-V$^+$ cells for individual mice (symbols) and means (bars) from five experiments are plotted normalized to the Ctrl mean. **d** Proportion of Annexin-V$^+$ B cells as a function of Prmt5 protein levels measured by WB and normalized to Actin 72 h post-stimulation. Spearman's correlation coefficient (*r*) and *p*-value (*p*) are indicated. **e** WB of Prmt5 and Revert protein staining as loading control in extracts of splenic B cells from Cγ1-cre (Ctrl) and *Prmt5$^{F/F}$* Cγ1-cre (F/F) mice, stimulated as in **a** for 72 h. The normalized proportion of Annexin-V$^+$ cells for individual mice (symbols) from three experiments and mean values (bars) are plotted. **f** Top, resting splenic B cells from *Prmt5$^{F/+}$* CD19-cre (Ctrl) or *Prmt5$^{F/F}$* CD19-cre (F/+) and *Prmt5$^{F/F}$* CD19-cre (F/F) mice plated onto 40LB cells with 1 ng/mL IL-4 to generate GC-like B cells (iGBs). Mean ± s.e.m. cell counts per day are plotted for two experiments with two mice each. Bottom, proportion of Annexin-V$^+$ iGBs for individual mice (symbols) and means (bars) for both experiments are plotted, pooling +/+ and F/+ as controls. **g** Cγ1-cre (Ctrl) and *Prmt5$^{F/F}$* Cγ1-cre (F/F) mice iGBs analyzed as in **f**. **h** Representative cell cycle profile as in **b** in Cγ1-cre (Ctrl) and *Prmt5$^{F/F}$* Cγ1-cre (F/F) iGBs at day 4. Means + s.d. of six mice per genotype from three experiments are plotted. Unpaired, two-tailed Student's *t* test was performed in **a**, **c**, **e–g**, only significant *p*-values are shown

**Prmt5 promotes activated B cell proliferation**. Ex vivo activation of purified resting B cells with mitogens supports limited proliferation (Fig. 3a). We therefore used a system that permits sustained B cell proliferation. We plated resting B cells onto 40LB feeder cells, expressing CD40L and the anti-apoptotic BAFF plus IL-4, which drives exponential proliferation and imparts a GC-like phenotype (hereafter iGBs)[33]. Prmt5 was the most expressed PRMT in wt iGBs (Supplementary Fig. 3A). *Prmt5$^{F/F}$* CD19-cre iGB cells showed only a small sDMA reduction by day 6 post-plating (Supplementary Fig. 3B), which limited the interpretation of the experiment. Nonetheless, these cells expanded poorly compared to controls, without increased apoptosis (Fig. 3f), suggesting both phenotypes could be separated. Indeed, expansion of *Prmt5$^{F/F}$* Cγ1-cre iGBs was compromised without any increase in apoptosis (Fig. 3g), despite efficient and sustained depletion of Prmt5 and sDMA (Supplementary Fig. 3C, 3D). Cell

cycle analysis of *Prmt5$^{F/F}$* Cγ1-cre iGBs revealed G1 arrest and reduced proportion of S-phase, which could explain reduced proliferation (Fig. 3h). We conclude that Prmt5 promotes B cell proliferation by a different mechanism than it protects B cells from apoptosis during activation.

**Prmt5 is essential for the antibody response**. To examine the function of Prmt5 in normal B cell proliferation in vivo without the confounding effect of apoptosis during activation, we analyzed GC responses in *Prmt5$^{F/F}$* Cγ1-cre mice immunized with NP-CGG. *Prmt5$^{F/F}$* Cγ1-cre mice showed >10-fold lower total anti-NP IgG1 than controls at day 14 post-immunization (Fig. 4a), which correlated with a severe deficit in antibody secreting cells (Fig. 4b). The drastic impairment of the antibody response in *Prmt5$^{F/F}$* Cγ1-cre mice was consistent at all times

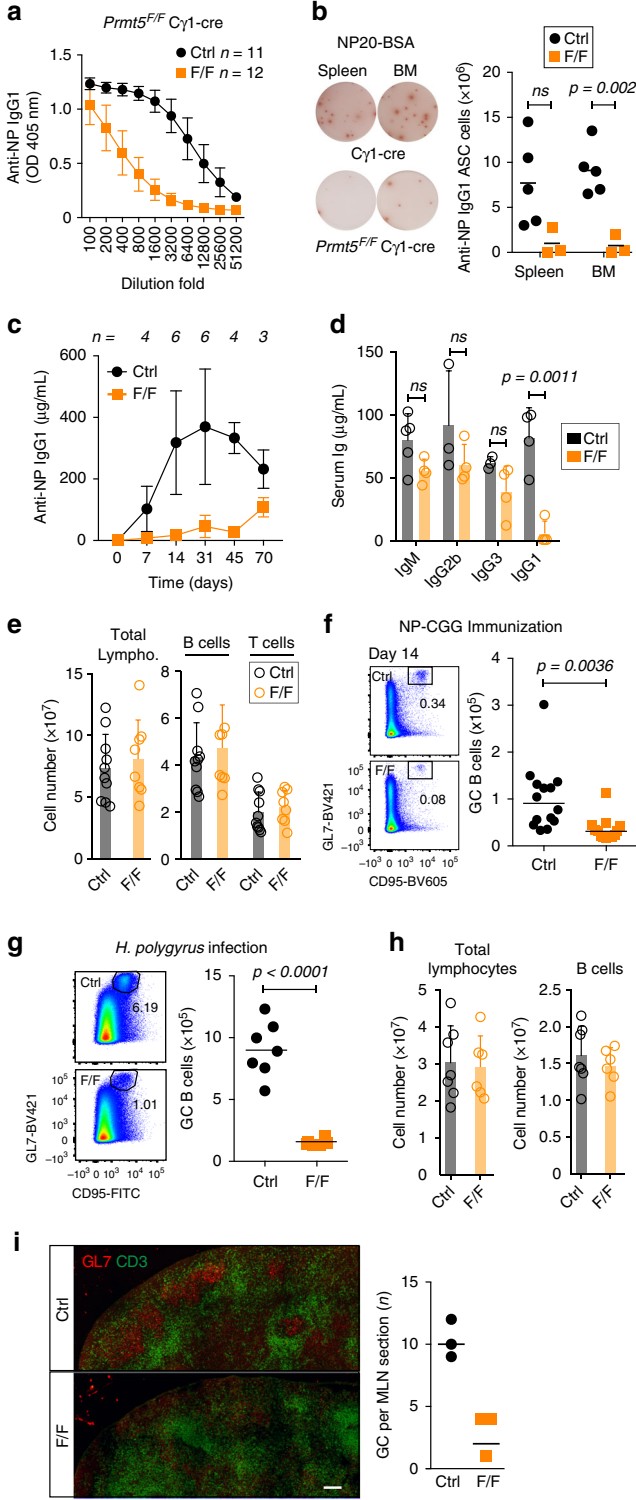

**Fig. 4** Antibody response and GC defects caused by Prmt5 deficiency. **a–i** Cγ1-cre (Ctrl) and *Prmt5^{F/F}* Cγ1-cre (F/F) mice were used throughout. **a** Total anti-NP IgG1 in the serum of mice, measured by ELISA 14 days after NP-CGG immunization. Mean ± s.d. OD values for serial dilutions are plotted for *n* mice from three experiments. **b** Representative pictures of ELISPOT for NP-specific IgG1 antibody secreting cells (ASC) at day 14 post-immunization. The number of ASC of individual mice (symbols) and means (bars) are plotted. **c** Anti-NP IgG1 in the serum of mice at various times post-immunization. Mean ± s.d. values for *n* mice at each time point from two experiments are plotted. **d** Total levels of antibody isotypes in the serum of *n* non-immunized mice. **e** Mean + s.d. number of lymphocytes per spleen at day 14 post-immunization, enumerated by flow cytometry for *n* mice from two experiments. **f** Representative flow cytometry plots (gated on B220+) of splenic GC B cell proportions at 14 days post-immunization with NP-CGG. The number of GC B cells per spleen for individual mice (symbol) and medians (bars) from three experiments are plotted. **g** As in **f**, for MLN of mice infected with *H. polygyrus* for 14 days. Data from two experiments are plotted. **h** Mean + s.d. number of lymphocytes per MLN in the mice from **g**. **i** Representative IF in MLN from mice infected with *H. polygyrus* from **g**, stained for the indicated antigens. Scale bar, 100 μm. GC numbers per MLN scored in individual mice (symbols) are plotted to the right. *p*-Values throughout are by an unpaired, two-tailed Student's *t* test

persistence cannot moderate the GC defect caused by Prmt5-deficiency. The number of total activated CD4+ T cells, Th2, or Treg cells was not significantly different between the groups (Supplementary Fig. 4A). In addition, there was no significant difference in the absolute number or ratio of Tfh/Tfr cells between the groups (Supplementary Fig. 4B). We conclude that, separate from its function during B cell activation, Prmt5 is essential for the antibody response by promoting GC formation, most likely through a B cell intrinsic function.

**Prmt5 is required for GC expansion and dynamics.** To pinpoint the GC stage at which Prmt5 played a role, we analyzed GC kinetics in *Prmt5^{F/F}* Cγ1-cre and Cγ1-cre control mice immunized with SRBC. Both groups showed similar number of GC B cells at day 5 post-immunization (Fig. 5a). The number and organization of GC appeared similar between the groups at this time (Fig. 5b, c), despite Prmt5 and sDMA being efficiently depleted in *Prmt5^{F/F}* Cγ1-cre GC (Fig. 5d). In contrast, GC B cell proportion and numbers were severely reduced by day 8 post-immunization in *Prmt5^{F/F}* Cγ1-cre mice (Fig. 5a, b), although residual GCs maintained overall organization (i.e. distinct DZ proximal to the T cell zone and CD35+ LZ) (Fig. 5c). Consistent with the ex vivo data, GC B cells in *Prmt5^{F/F}* Cγ1-cre mice did not show apoptosis (Fig. 5e) but had a larger proportion of Ki67^low cells (Fig. 5f), indicative of reduced proliferation[34].

To better understand the GC defect, we examined DZ and LZ B cells in SRBC-immunized *Prmt5^{F/F}* Cγ1-cre mice. Despite an apparently normal DZ/LZ ratio, *Prmt5^{F/F}* Cγ1-cre mice immunized with SRBC or infected with *H. polygyrus* displayed an atypical Cxcr4−Cd86^low GC population (Fig. 5g, h). A large proportion of Cxcr4−Cd86^low cells were Ki67^low (Fig. 5i). This population was not caused by any potential defect in affinity maturation, as it was also observed in AID-deficient mice (Fig. 5j), and did not express AID, as shown using *Prmt5^{F/F}* Cγ1-cre *Aicda-GFP* mice (Fig. 5k). The latter mice revealed further alterations in Prmt5-null GC B cells, including the loss of an AID^dim DZ population and fewer LZ AID+ cells, presumably because they converted to the Cxcr4−Cd86^low subset (Fig. 5k).

We conclude that Prmt5 is necessary for GC expansion by promoting cell proliferation and normal GC dynamics.

tested (Fig. 4c). Furthermore, *Prmt5^{F/F}* Cγ1-cre mice showed a 17-fold reduction in pre-immune serum IgG1, which is elicited against environmental antigens, indicating that chronic stimulation could not compensate for the defect (Fig. 4d). The resting splenic B cell populations were normal in *Prmt5^{F/F}* Cγ1-cre mice, but GC B cell numbers were reduced threefold at day 14 post-immunization (Fig. 4e, f). Accordingly, *Prmt5^{F/F}* Cγ1-cre mice chronically infected with *H. polygyrus* exhibited smaller and fewer GC, with sixfold lower GC B cells in MLN, while total B and T cell numbers were unaffected (Fig. 4g–i). Thus, antigen

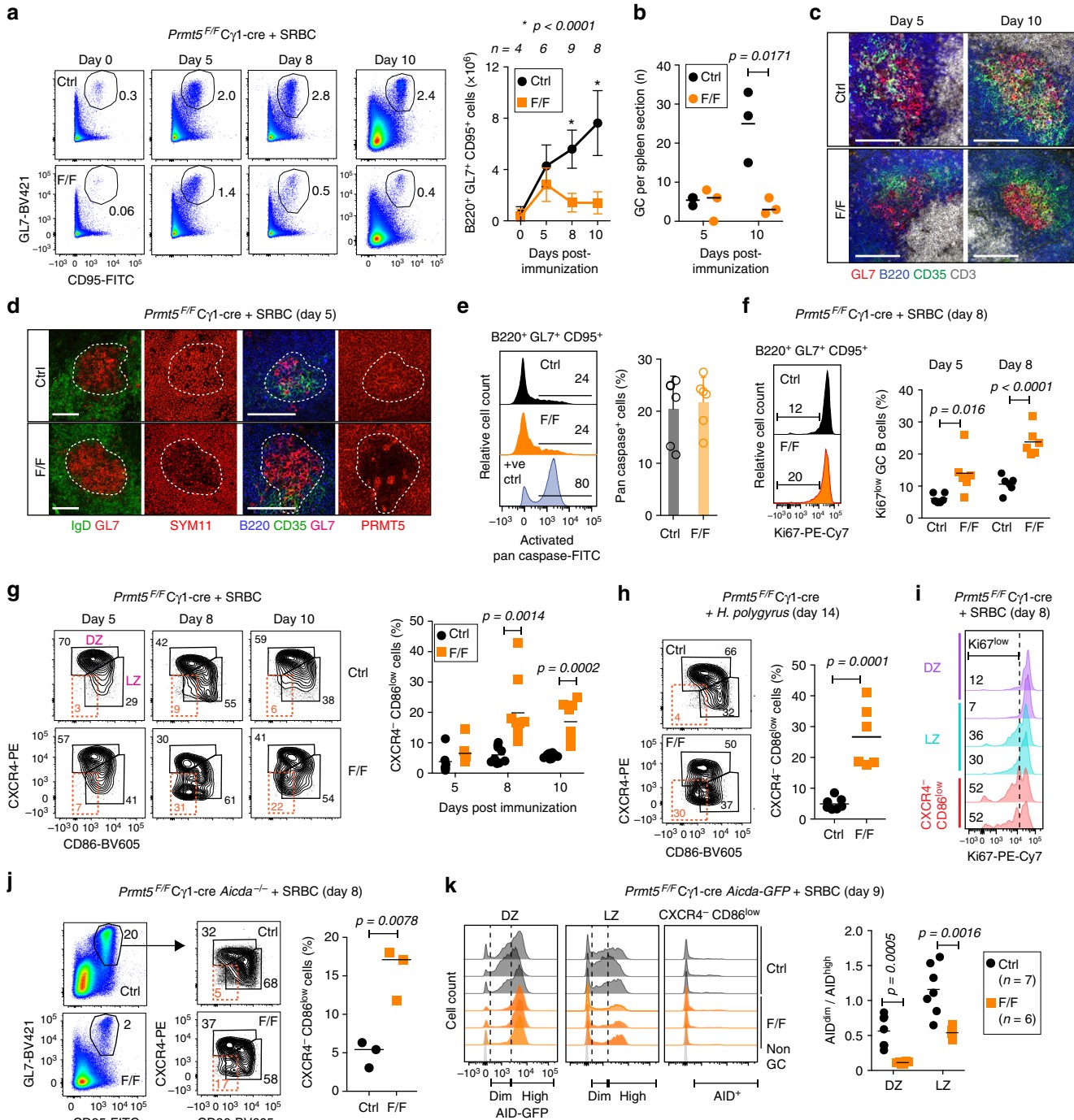

**Fig. 5** Prmt5 is necessary for GC expansion. **a–l** Cγ1-cre (Ctrl) versus *Prmt5^F/F* Cγ1-cre (F/F) mice. **a** Representative flow cytometry plots for GC B cells at various times post-SRBC immunization. Mean ± s.d. absolute GC B cell numbers from *n* mice from two to three experiments are plotted. **b** GCs per spleen section in individual mice (symbols), with means (bars). **c** Representative fluorescent microscopy images of spleen sections from mice in **a** stained for B cells (B220), T cells (CD3), FDCs (CD35), and activated B cells (GL7). Scale bar, 100 μm. **d** Representative IF confocal images in splenic sections from immunized mice, stained for the indicated markers, Prmt5 and sDMA (SYM11). GCs are contoured. Scale bars, 100 μm. **e** Representative histograms of activated pan-caspase staining in splenic GC B cells at day 8 post-SRBC immunization. Etoposide (3 μM) was used to induce apoptosis as a positive control. Mean + s.d. proportion of caspase+ GC B cells for six mice per group from two independent experiments are plotted. **f** Representative histograms of Ki67 expression in splenic GC B cells. Mean + s.d. proportion of Ki67^low GC B cells for individual mice (symbols) from two experiments and means (bars) are plotted. **g** Representative flow cytometry plots of dark zone (DZ), light zone (LZ), and a CXCR4− CD86^low GC B cells (gated on B220+ Gl7+ CD95+) in SRBC-immunized mice. The proportion of CXCR4− CD86^low GC B cells for individual mice (symbols) from two to three experiments and means (bars) are plotted. **h** GC B cells in mice infected with *H. polygyrus* for 14 days analyzed as in **g**. Data from two experiments. **i** Representative histograms of Ki67 levels in GC B cell subsets in mice immunized with SRBC. **j** GC B cells and subpopulations as in **g** for *Aicda^−/−* Cγ1-cre (Ctrl) or *Aicda^−/− Prmt5^F/F* Cγ1-cre (F/F) mice immunized with SRBC. **k** Histograms of AID-GFP levels for individual *Aicda*-GFPtg Cγ1-cre (Ctrl) or *Aicda*-GFPtg *Prmt5^F/F* Cγ1-cre (F/F) mice at day 9 post-SRBC immunization. The AID^dim/AID^high ratio for individual mice (symbols) from two experiments and means (bars) are plotted. *p*-Values indicated throughout are by an unpaired, two-tailed Student's *t* test

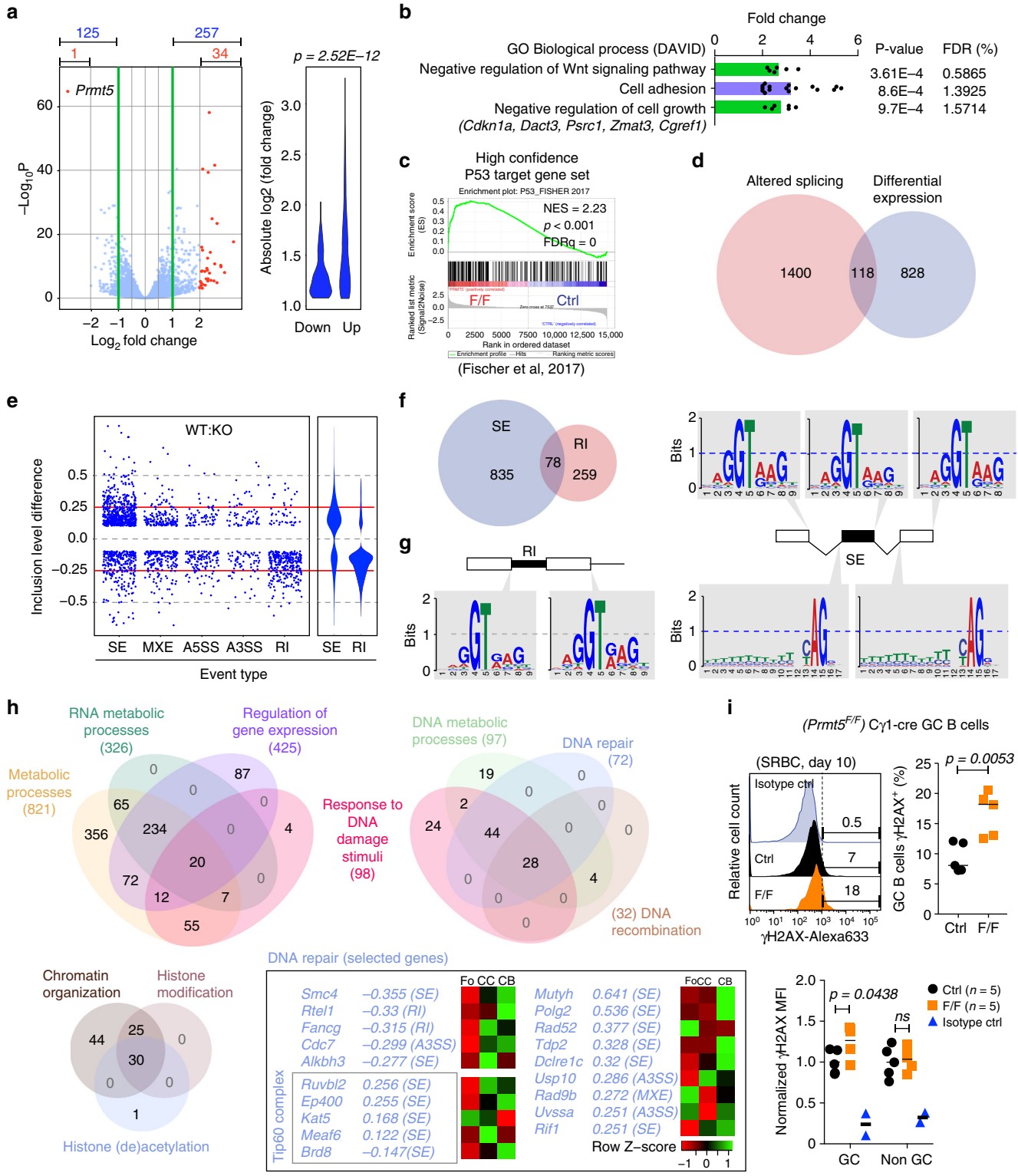

**Prmt5 regulates B cell transcription and splicing fidelity.** We performed RNA-seq for mechanistic insight into the defects of Prmt5-deficient B cells. Given the paucity of GC B cells in vivo, we used $Prmt5^{F/F}$ Cγ1-cre and Cγ1-cre iGBs at day 4 after plating, when Prmt5 and sDMA depletion are complete and iGBs are $CD95^+$ $GL7^+$ (Supplementary Figs. 3C, 3D, 5A). Prmt5 ablation induced large transcriptional changes, with 1511 genes differentially expressed in Prmt5-null versus control iGBs (p-adj < 0.05, ≥1.5-fold change in either direction) (Supplementary Data 1).

Focusing on genes that changed by ≥2-fold reduced the complexity to 382 genes, with a majority being upregulated and displaying larger changes (Fig. 6a), in line with the predominant role of Prmt5 as transcriptional repressor[18,21].

To identify biological processes affected by Prmt5 deficiency, we performed functional annotation by gene ontology (GO). No significantly enriched (FDR <5%) GO term was found amongst the downregulated genes, regardless of the fold change in expression (see Methods). On the other hand, analyzing all

**Fig. 6** Prmt5 regulates gene expression and maintains splicing fidelity in B cells. **a** Volcano plot of gene expression changes in $Prmt5^{F/F}$ Cγ1-cre versus Cγ1-cre iGB cells. The number of genes significantly changed by ≥2- or ≥4-fold (in red), are indicated. The violin plot shows the absolute change for the genes changing by ≥2-fold, p value from two-sided $t$-test. **b** GO terms significantly enriched for the 257 genes upregulated ≥2-fold in Prmt5-null iGBs, analyzed against a background list of all expressed genes in iGBs at the DAVID server. **c** GSEA of a curated set of 346 genes that are direct targets of p53 and are upregulated in response to p53. **d** Venn diagrams comparing genes differentially expressed by ≥1.5-fold to genes with at least one splicing alteration. Only significant events ($P$-value <0.05) for well-expressed genes (basemean >50) are included. **e** Scatter plot of inclusion level difference (ILD) in WT/KO for each significant splicing event ($P < 0.05$, ≥10% ILD) separated by category (SE, skipped exon; MXE, mutually exclusive exons; A5SS and A3SS, altered 5′ and 3′ splicing site; RI, retained intron). Violin plots for SE and RI highlight the preferential direction of the change in each. **f** Venn diagram of significant skipped exon (SE) and retained intron (RI) events. **g** Sequence logos of the splicing donor and acceptor sites around RI and SE events in $Prmt5^{F/F}$ Cγ1-cre iGBs. Each letter's height represents the probability of appearing at that position. **h** Venn diagrams of chosen overlapping GO terms containing genes with significantly affected splicing in Prmt5-null B cells. Selected genes involved in DNA repair are indicated along with ILD and splicing event type. **i** Representative histograms of γH2AX staining in Cγ1-cre (Ctrl) and $Prmt5^{F/F}$ Cγ1-cre (F/F) mice GC B cells (B220$^+$ Gl7$^+$ CD95$^+$), 10 days post-SRBC immunization. Proportion of γH2AX$^+$ and normalized γH2AX levels estimated from mean fluorescent intensity (MFI) are plotted for five mice (symbols) per genotype from two experiments. $p$-Values are by an unpaired, two-tailed Student's $t$ test

upregulated genes ranked by fold increase identified many GO terms related to cell adhesion and extracellular sensing, as well as signaling and proliferation (Supplementary Figs. 5B, Supplementary Data 2). Restricting the GO analysis to the 257 genes upregulated by ≥2-fold also showed enrichment in cell adhesion and negative regulators of cell proliferation that included several p53 target genes (Fig. 6b). Gene set enrichment analysis (GSEA) confirmed the upregulation of adhesion and cytokine signaling signatures, but revealed the upregulation of a p53 transcriptional signature, including pro-apoptotic genes (*Perp*, *Bbc3*, *Tnfrsf10b*, *Bax*, *El24*, etc) and the cell cycle inhibitor *Cdkn1a* in Prmt5-null B cells (Supplementary Fig. 5C and Supplementary Data 2). Analysis of a curated set of 346 genes upregulated by p53[35] confirmed a significant upregulation of p53 target genes in Prmt5-deffient iGBs, (Fig. 6c).

Prmt5 regulates splicing[25,36]. RNA-seq data analysis revealed significant alterations (by >10%) in the splicing of 1518 genes in Prmt5-null B cells (Supplementary Data 3). Genes affected at the splicing and transcriptional level showed little overlap (Fig. 6d), indicating that aberrant splicing was not directly responsible for the gene expression changes in Prmt5-null B cells. There was a clear preference to exclude exons (SE events) and include introns (RI events) in Prmt5-deficient cells (Fig. 6e), but most genes affected by SE or RI did not overlap (Fig. 6f). Both types of event had the same underlying cause; the skipping of weak 5′ splicing donor sites in Prmt5-deffient B cells, while 3′ acceptor site choice was not affected (Fig. 6g). Two major SE events resulted in exon exclusions affecting a large proportion of the Mdm4 transcripts (Supplementary Fig. 5D). These Mdm4 forms cannot repress the transcriptional activity of p53[25], which likely contributes to the p53 gene expression signature in Prmt5-deficient B cells. Functional annotation of genes with significantly affected splicing, considering either all events or just those with ≥0.25 inclusion level difference, revealed a predominance of GO terms related to regulation of gene expression and RNA processing (Supplementary Data 4). Splicing of multiple genes encoding chromatin and histone modification factors was affected including transcripts of several subunits of the Tip60 complex, *Hdac6* and *Chd1l* (Fig. 6h, Supplementary Fig. 5E), which regulate the p53 transcriptional response but also participate in DNA repair[37,38]. Furthermore, splicing of transcripts encoding multiple DNA damage response and repair genes from the base excision, Fanconi and homologous recombination repair pathways, which are induced in GC B cells, was affected (Fig. 6h). Accordingly, the DNA damage marker γH2AX was significantly increased in Prmt5-null iGBs (Supplementary Fig. 5F), as well as in GC B cells of immunized $Prmt5^{F/F}$ Cγ1-cre mice (Fig. 6i). AID-induced DNA damage did not cause the GC defects, as ablating AID did not rescue GC expansion or normal populations

in $Aicda^{-/-}$ $Prmt5^{F/F}$ Cγ1-cre mice (Fig. 5j, Supplementary Fig. 5G).

We conclude that Prmt5 maintains the normal GC transcriptional program and protects from spontaneous DNA damage, at least in part by enforcing the use of weak 5′ donor sites to regulate proper RNA processing of multiple chromatin modifiers and DNA repair factors.

**A p53 response eliminates Prmt5-deficient Pro-B cells.** Based on the RNA-seq analysis and the relevance of the p53 response in other Prmt5-deficient cell types[25,39–41], we tested the contribution of the p53 response to the phenotype of Prmt5-deficient B cells in vivo. Ablation of Prmt5 in Pro-B cells in $Prmt5^{F/F}$ Mb1-cre mice[29] resulted in the absence of mature B cells in the spleen (Fig. 7a). This was due to a block of B cell development at the Pro-B cell stage (Hardy's fraction B) (Fig. 7b). Progenitor cells initiate VDJ recombination, which evokes a p53 apoptotic response that limits the number of Pro-B cells in vivo[42,43]. Introducing the pre-rearranged B1-8 V$_H$ allele in $Prmt5^{F/F}$ Mb1-cre mice failed to rescue Pro-B cells (Fig. 7c). On the other hand, $Prmt5^{F/F}$ Mb1-cre $Tpr53^{-/-}$ mice, fully recovered Pro-B cell numbers and substantially recovered fraction C/C′, corresponding to late Pro- and early Pre-B cell stages, but not the late Pre-B cell fraction D (Fig. 7d). Closer examination of fraction C/C′ showed the lack of Pre-BCR$^+$ cells (fraction C′) in $Prmt5^{F/F}$ Mb1-cre $Tpr53^{-/-}$ mice (Fig. 7e). We conclude that Prmt5 is essential for B cell development by dampening the p53 response in Pro-B cells and by another essential but p53-independent function in Pre-B cells, but we cannot rule out a p53-independent function of Prmt5 in VDJ-recombination.

**Trp53-independent defects in Prmt5-null B cells.** We asked whether a p53 response might explain the apoptosis observed when Prmt5-null B cells were activated. Both p53 protein levels and activating phosphorylation at Ser15 were increased in $Prmt5^{F/F}$ CD19-cre B cells activated ex vivo (Fig. 8a). Accordingly, the p53 targets *Bax* and *Cdkn1a* were upregulated in these cells (Fig. 8b). Nevertheless, ablating p53 did not prevent the increase in apoptosis or reduced proliferation, as shown in B cells from $Prmt5^{F/F}$ CD19-cre $Trp53^{-/-}$ mice (Fig. 8c).

$Prmt5^{F/F}$ Cγ1-cre iGB cells do not undergo apoptosis but still fail to expand and show a p53 transcriptional response with *Cdkn1a* (encoding p21) upregulation (Figs. 3g, 6c, Supplementary Fig. 5C). We asked whether the proliferation defect observed in Prmt5-deficient B cells was p53- and/or p21-dependent. Pharmacological inhibition of Prmt5 similarly reduced the proliferation of wt, $Trp53^{-/-}$ and $Cdkn1a^{-/-}$ iGB cells, without any noticeable difference in their sensitivity to the inhibitor

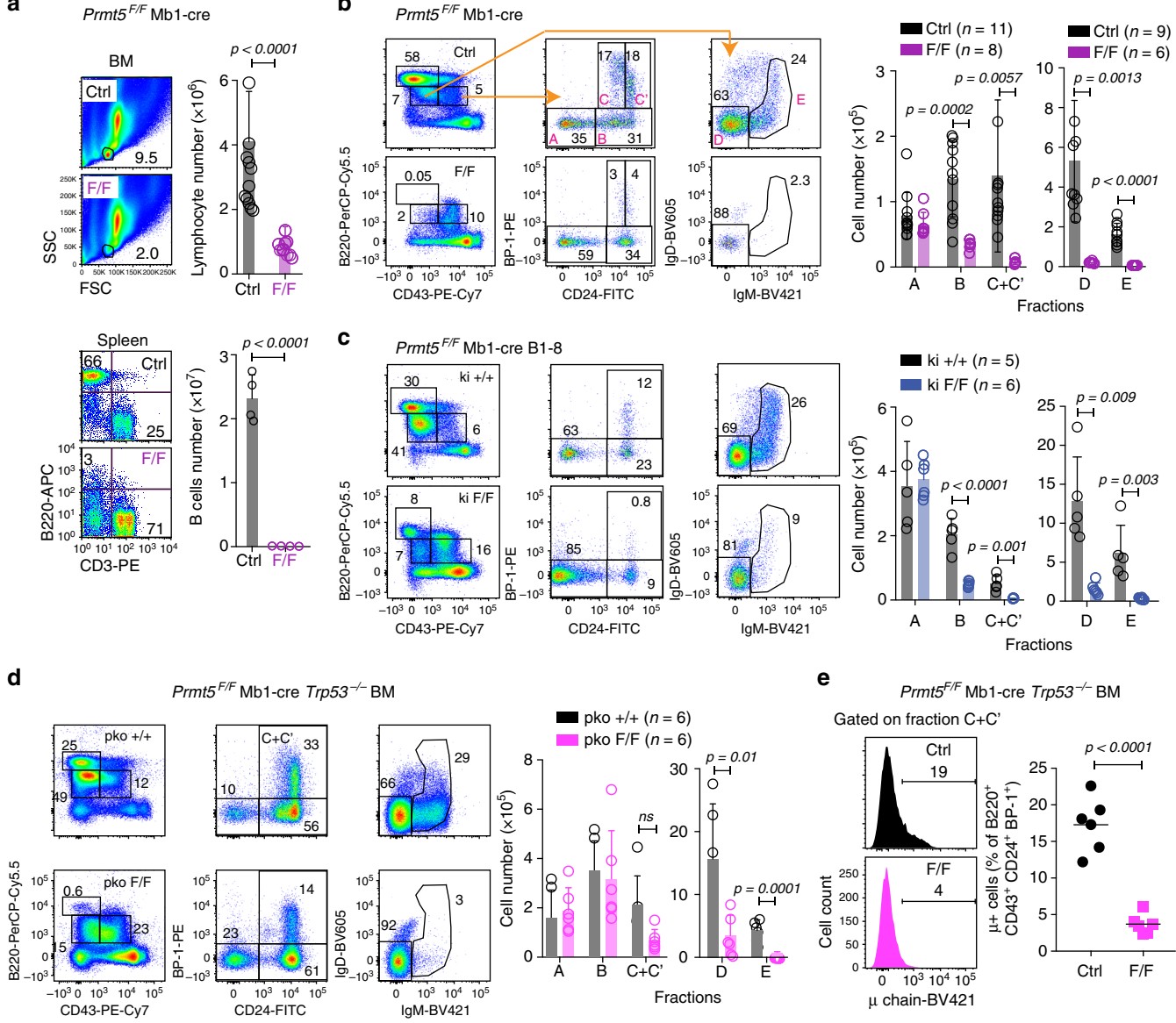

**Fig. 7** Prmt5 is required for B cell development. **a** BM lymphocytes and splenic B cells in MB1-cre (Ctrl) or *Prmt5^F/F* MB1-cre (F/F) mice. Representative flow cytometry plots and bar plots of mean + s.d. absolute lymphocyte counts for *n* mice from four experiments. **b** Representative flow cytometry plots gating for Hardy fractions A to E in BM from 3–4-months-old MB1-cre (Ctrl) and *Prmt5^F/F* MB1-cre (F/F) mice. Means + s.d. of absolute cell numbers for each fraction from *n* of mice from four independent experiments are plotted. **c** As in **b**, for MB1-cre B1-8 (ki+/+) and *Prmt5^F/F* MB1-cre B1-8 (ki F/F) mice, from three experiments. **d** As in **b**, for MB1-cre *Trp53^−/−* (pko+/+) and *Prmt5^F/F* MB1-cre *Trp53^−/−* (pko F/F) mice, from three experiments. **e** Proportion of large Pre-B cells in fractions C+C' estimated by the pre-BCR expression in individual (symbols) MB1-cre *Trp53^−/−* (Ctrl) and *Prmt5^F/F* MB1-cre *Trp53^−/−* (F/F) mice. *p*-Values throughout are from an unpaired, two-tailed Student's *t* test

(Fig. 8d–f). To genetically confirm this, we derived iGBs from *Prmt5^F/F* Cγ1-cre *Trp53^−/−* mice, which showed similarly reduced expansion and cell cycle defects than *Prmt5^F/F* Cγ1-cre iGBs (Fig. 8g, h), demonstrating that these phenotypes were p53-independent. Furthermore, immunized *Prmt5^F/F* Cγ1-cre *Trp53^−/−* mice displayed the same GC defects than *Prmt5^F/F* Cγ1-cre mice, including the accumulation of Cxcr4^− Cd86^low cells (Fig. 8i, j). Since *Cdkn1a* is not expressed in p53-deficient cells (Fig. 8f), these results also ruled out p21 as a cause for the defects in B cell expansion in vitro or GC in vivo.

We conclude that Prmt5 prevents apoptosis upon B cell activation independently of p53 activation, and that it has additional p53- and p21-independent functions that promote B cell proliferation and the GC reaction.

**Prmt5 functions in LZ B cells**. We then analyzed the expression of Prmt5 in GC B cell subsets. Gene expression data indicated higher Prmt5 mRNA levels in centrocytes than centroblasts (Supplementary Fig. 6A). Moreover, IHC suggested polarization of Prmt5 expression in the GC (Fig. 9a, Supplementary Fig. 6B). IF with the LZ marker CD35 confirmed that Prmt5 protein was preferentially expressed in the LZ GC B cells (Fig. 9b). In line with these results, Prmt5-deficient iGBs were significantly depleted of gene expression signatures characteristic of CXCR4^− LZ B cells[6,44], as well as genes upregulated by CD40 engagement, while DZ-related gene sets were not significantly affected (FDR >5%) (Fig. 9c, Supplementary Fig. 6C). To obtain an insight into the dynamics of *Prmt5^F/F* Cγ1-cre GC B cells, we analyzed the available gene expression datasets from

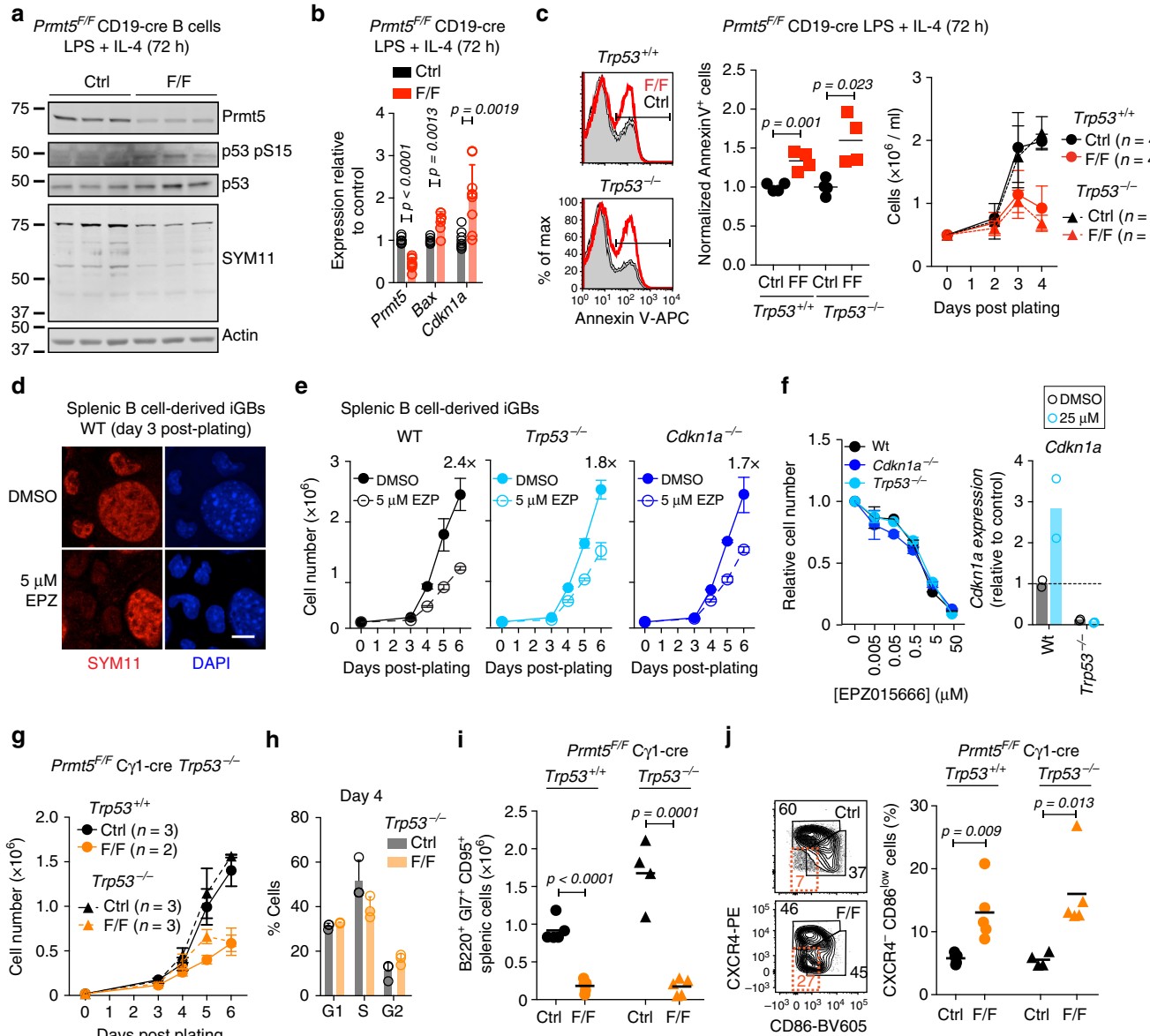

**Fig. 8** Prmt5 loss induces p53-independant apoptosis. **a** WB probed for the indicated proteins on extracts from CD19-cre (Ctrl) and *Prmt5*[F/F] CD19-cre (F/F) splenic B cells cultured with LPS and IL-4 for 3 days. **b** Gene transcript levels by RT-qPCR in splenic B cells from **a**. Mean + s.d. RNA level normalized to Actin for *n* mice are plotted relative to the Ctrl mean. **c** Representative histograms of Annexin-V levels in CD19-cre (Ctrl) and *Prmt5*[F/F] CD19-cre (F/F) mice in either *Trp53*[+/+] or *Trp53*[−/−] background. Annexin-V[+] proportion for individual mice and mean + s.e.m cell concentration over time are plotted from one experiment. **d** Representative confocal microscopy of sDMA (SYM11) IF in iGB cells from wt mice treated with DMSO or 5 μM EPZ for 48 h. Large nuclei are from feeder cells. Scale bar, 10 μm. **e** Expansion of iGBs derived from wt, Trp53[−/−] and *Cdkn1a*[−/−] splenic B cells, treated with DMSO or EPZ 24 h after plating. Means ± s.e.m cell counts of two mice from one experiment are plotted. **f** Sensitivity of iGB cells to EPZ 4 days after treating with EPZ doses. Relative mean ± s.e.m cell number and *Cdkn1a* expression by RT-qPCR are plotted for two mice from one experiment. **g** Expansion of iGBs from Cγ1-cre (Ctrl) and *Prmt5*[F/F] Cγ1-cre (F/F) mice and their *Trp53*[−/−] counterparts. Mean ± s.e.m of cell count for *n* mice are plotted. **h** Cell cycle profile of Cγ1-cre *Trp53*[−/−] (Ctrl) and *Prmt5*[F/F] Cγ1-cre *Trp53*[−/−] (F/F) iGBs pulsed with BrdU for 1 h at day 4 and stained with anti-BrdU and propidium iodide (PI). Means + s.d. for three mice per genotype from two independent experiments are plotted. **i** GC B cells in the spleen of Cγ1-cre *Trp53*[−/−] (Ctrl) and *Prmt5*[F/F] Cγ1-cre *Trp53*[−/−] (F/F) mice 10 days after immunization with SRBC. Individual mice (symbols) and mean (bars) values are plotted from three experiments. **j** Representative flow cytometry and proportion of GC Cxcr4− Cd86[low] B cells in the mice in **i**. *p*-Values throughout are from an unpaired, two-tailed Student's *t* test

functional GC B cell subsets. We found that Prm5 was preferentially expressed in positively selected centrocytes, defined either by their transient c-Myc expression[7] or increased antigen presentation ability[45] (Fig. 9d). Accordingly, the gene signatures of Myc[+] GC B cells and mTORC signaling associated to positive selection were significantly downregulated in *Prmt5*-null B cells (Fig. 9c).

We conclude that Prmt5 is regulated to favor expression in the LZ and that it contributes to maintain the transcriptional identity of LZ B cells.

**Prmt5 prevents premature plasma cell differentiation.** Additional gene expression analyses suggested that Prmt5-null B cells were prone to plasma cell differentiation. Prmt5 was

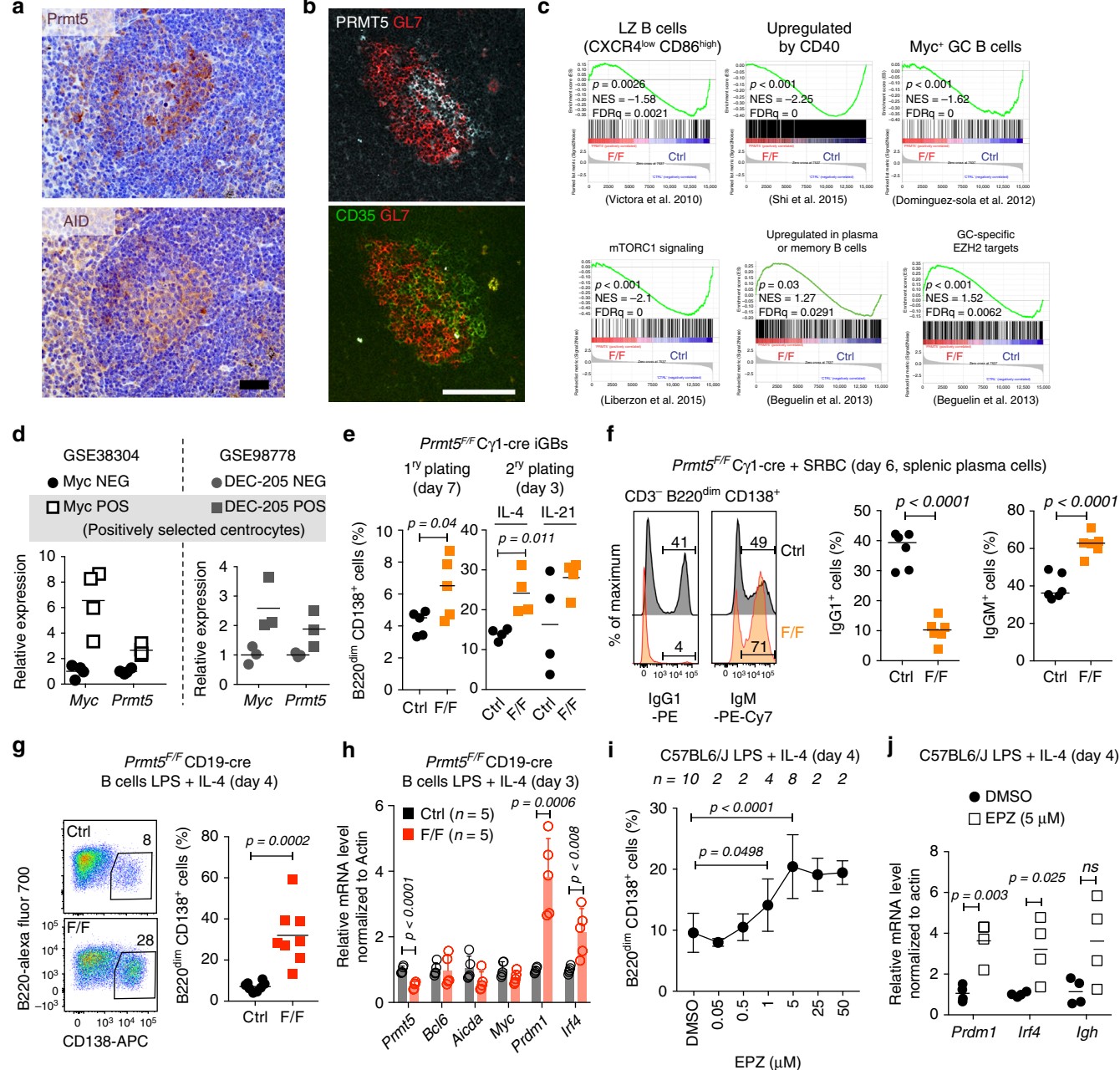

**Fig. 9** Prmt5 acts in the light zone to prevent B cell differentiation. **a** IHC images for Prmt5 and AID, as GC marker, on consecutive spleen sections from wt mice 14 days post-NP-CGG immunization. Representative of three mice from two experiments. Scale bar, 40 μm. **b** Representative IF on wt mouse spleen section 10 days after SRBC immunization, stained for GL7 CD35 (LZ marker) and Prmt5. Scale bar, 100 μm. **c** GSEA of transcriptional changes in *Prmt5*[F/F] Cγ1-cre (F/F) versus Cγ1-cre (Ctrl) iGB cells against the indicated gene signatures. Enrichment was considered significant for P < 0.05 and FDR < 5%. **d** Relative gene expression in positively selected GC B cells. **e** Proportion of plasma cell-like B cells in Cγ1-cre (Ctrl) or *Prmt5*[F/F] Cγ1-cre (F/F) iGB cultures at day 7 post-plating, or at day 3 after replating with 1 ng/mL IL-4 or 10 ng/mL IL-21. Individual mice (symbols) from one experiment are plotted. **f** Representative flow cytometry histograms of intracellular IgG1 and IgM staining of plasma cells in *Prmt5*[F/F] Cγ1-cre (F/F) or Cγ1-cre (Ctrl) mice 6 days post-SRBC immunization. Proportion of IgG1[+] and IgM[+] plasma cells in individual mice (symbols) from two experiments are plotted. **g** Representative flow cytometry plots of plasma cells in CD19-cre (Ctrl) or *Prmt5*[F/F] CD19-cre (F/F) B cells cultured 4 days in LPS and IL-4. Plasma cell proportion for individual mice (symbols) from three independent experiments are plotted. **h** Gene expression by RT-qPCR in splenic B cells cultured as in **g** for 3 days. Mean + s.d. mRNA levels normalized to Actin for n mice from two experiments are plotted relative to Ctrl average. **i** Mean ± s.d. (or ± s.e.m for n = 2) of plasma cell proportion in splenic B cells stimulated with LPS and IL-4 and treated with various EPZ doses for 4 days, are plotted for n mice from one to five experiments. **j** Gene expression by RT-qPCR in B cells treated with 5 μM EPZ as in **i**; individual mice (symbols) are plotted. Bars indicate mean values in all scatter plots. Significant p-values throughout are by an unpaired, two-tailed Student's t test except for GSEA, in which statistics are indicated

preferentially expressed in the Myc$^+$ and Irf4$^+$ subset of LZ B cells recently shown to harbor plasmablast precursors[8] (Supplementary Fig. 6D). We also noticed that many cell adhesion and surface markers overexpressed in Prmt5-null iGBs are upregulated in memory and/or plasma cells relative to GC B cells (Supplementary Fig. 6E). Furthermore, a subset of the genes defining a composite memory/plasma cells signature, as well as some of the genes normally repressed by EZH2 in GC B cells[46], were upregulated in Prmt5-null iGBs (Fig. 9c). These enrichments were modest, fitting with the observation that the *Prmt5$^{F/F}$* Cγ1-cre iGBs at day 4 used for RNA-seq showed no increase in CD138$^+$ cells, had normal Bcl6 levels, and no change in Bcl6 target genes expression (Supplementary Figs. 5A and 6C). Nonetheless, by day 7, or after replating, *Prmt5$^{F/F}$* Cγ1-cre iGB cells showed a greater proportion of CD138$^+$ cells and reduced proliferation potential than the control (Fig. 9e and Supplementary Fig. 6F), as expected for differentiated cells. Moreover, *Prmt5$^{F/F}$* Cγ1-cre mice immunized with SRBC showed a significantly lower proportion of IgG1$^+$ and a concomitant increase in IgM$^+$ plasma cells at day 6 (Fig. 9f, Supplementary Fig. 6H), when most plasma cells are of extrafollicular origin. These results in vivo are consistent with premature plasma cell differentiation, before isotype switching can take place.

To further test whether Prmt5-null B cells were intrinsically predisposed to differentiation, we stimulated splenic B cells ex vivo with LPS + IL-4, a model of plasma cell differentiation[5]. Because of the slow kinetics of excision in *Prmt5$^{F/F}$* Cγ1-cre B cells in these conditions, we used B cells from *Prmt5$^{F/F}$* CD19-cre mice, which also yielded more CD138$^+$ cells and upregulated *Prdm1* and *Irf4* compared to control cells, irrespective of their p53 status (Fig. 9g, h, Supplementary Fig. 6G). Importantly, Prmt5 deficiency had little effect on intrinsic class switch recombination per cell division (Supplementary Fig. 6I), ruling out this explanation for the reduced proportion of IgG1$^+$ plasma cells found in vivo. Finally, pharmacological inhibition of Prmt5 in wt B cells plated in LPS + IL-4 also caused a dose-dependent, but p21-independent, increase in CD138$^+$ cells, as well as *Prdm1*, *Irf4*, and *Igh* expression (Fig. 9i, j, Supplementary Figs. 6J, 6K).

We conclude that Prmt5 prevents premature plasma cell differentiation, which can partly explain the defect in GC expansion in vivo in mice with Prmt5-null B cells.

## Discussion

We demonstrate that Prmt5 is essential for B cell development, the GC reaction, and antibody responses. Prmt5 ensures cell survival, proliferation, and/or differentiation at critical B cell stage transitions, strongly influencing GC dynamics and output. The effect of ablating Prmt5 in B cells is pleiotropic, as expected for an enzyme that modifies a large number of substrates[24,47]. Nonetheless, we delineate distinct functions for Prmt5 in different B cells stages, the majority of which are novel p53-independent functions.

A partial block in B cell development but increased mature B cells had been reported after deletion of Prmt5 in all hematopoietic cells[48]. We find no mature B cells in *Prmt5$^{F/F}$* Mb1-cre mice demonstrating that Prmt5 is essential for B cell development in the BM. Prmt5 functions at least in Pro-B and large Pre-B cells, which are highly proliferative stages[1]. Pro-B cells are especially sensitive to p53-dependent apoptosis[42,43], and were rescued in *Prmt5$^{F/F}$* Mb1-cre *Trp53$^{-/-}$* mice. Thus, Prmt5 likely prevents a pro-apoptotic p53 response to allow Pro-B cell formation. B cell development was subsequently blocked before the large Pre-B cell stage, indicating another, p53-independent function of Prmt5 required for Pre-B cell formation or survival that remains to be investigated. While introducing a rearranged *Igh* failed to rescue

Pro-B cells, we cannot rule out a p53-independent function of Prmt5 in VDJ-recombination to explain the Pre-B cell block. However, the Prmt5 requirement for B cell development contrasts with the dispensable role of Prmt5 in T cell development[49]. Indirect regulation of cytokine signaling by Prmt5 via splicing (Supplementary Fig. 5C), as shown in T cells[49], and/or regulation of pre-BCR signaling, as shown for Prmt1[12], may contribute to this dependency.

All major functions of Prmt5 in mature B cells are p53-independent, in contrast to most other systems in which growth and survival defects associated to Prmt5 ablation are, to a large extent, p53-dependent[25,39–41]. Apoptosis upon activating B cells that were previously depleted of Prmt5-catalyzed sDMA is p53-independent, as shown by combining both deficiencies. To our knowledge, p53-independent apoptosis had only been described in Prmt5-deficient glioblastoma[50]. This mechanism is relevant in vivo, upon polyclonal B cell activation by alum or parasite infection. Identifying sDMA-modified proteins in resting B cells will help elucidating the mechanism, which acts soon after activation. Indeed, apoptosis in vivo is observed before the B cells show surface GL7 while deleting Prmt5 after B cell activation does not cause apoptosis.

Even after bypassing apoptosis, proliferation of Prmt5-null B cells remains compromised, revealing another function of Prmt5 in mature B cells. This proliferation defect is p53-independent, indicating that p53 activation is secondary to another defect in Prmt5-null B cells. In Prmt5-deficient mouse neural progenitor and human B cell lymphoma cells, p53 activation is partly caused by altered splicing producing an unstable form of the negative p53 regulator Mdm4[19,25], which we observe in Prmt5-null B cells but does not explain the p53-independent effects. Prmt5-null B cells accumulate DNA damage, as also observed in *C. elegans*, mouse primordial germ cells and human cancer cell lines[39,40,51], which could reflect reduced DNA repair. Prmt5 promotes homologous recombination repair by methylating RuvBl1 to enhance Tip60/Kat5 activity[28], and by regulating alternative splicing of *Kat5* in mouse fetal liver hematopoietic cells[52]. We find splicing fidelity defects in Prmt5-null B cells affecting transcripts of chromatin modifiers that regulate DNA repair, including Tip60, as well as multiple DNA repair factors. Nonetheless, additional splicing alteration, such as in cytokine signaling factors like *Il2rg* and *Jak3* just described in T cells[49] that we also found (Supplemental Data 3), are likely to compound in the reduced proliferation of Prmt5-null B cells in vivo.

Prmt5's polarized expression and defects caused by its ablation in GC dynamics and gene expression profile indicate a major function in LZ B cells. In line with the preferential expression of Prmt5 in the LZ B cell subsets undergoing positive selection and containing plasma cell precursors, our data are consistent with Prmt5 driving GC dynamics and preventing plasma cell differentiation. Premature differentiation of Prmt5-null B cells is supported by the upregulation of CD138 (syndecan), *Prdm1*, *Irf4* and *Igh* in B cells activated ex vivo and the increased IgM$^+$ to IgG1$^+$ ratio of splenic plasma cells in immunized *Prmt5$^{F/F}$* Cγ1-cre mice. These mice still show a deficit of ASC, suggesting an additional function of Prmt5 required in plasma cells in vivo. In other cell types, the role of Prmt5 in differentiation is context dependent[24]. Prmt5 deficiency causes upregulation of differentiation markers in ES cells[53] and promotes myeloid differentiation in leukemia cells driven by MLL-fusions[54]. Our data indicate that Prmt5 represses at least part of the plasma cell program. Thus, *Prmt5$^{F/F}$* Cγ1-cre iGBs from day 4 show a modest enrichment in transcriptional changes associated with plasma cell differentiation, such as upregulation of some EZH2 targets and surface markers, preceding the increase in CD138$^+$ cells observable at later timepoints. In addition, Prmt5 underpins GC B cell

proliferation, with null cells downregulating the Myc and mTOR gene expression signatures associated to DZ reentry[7,45]. The proportion of DZ and LZ B cells is reduced in $Prmt5^{F/F}$ Cγ1-cre mice because of an atypical population of Cxcr4− CD86[low] AID− B cells with low proliferation potential. The origin of these cells is unclear. We cannot rule out the relative accumulation of a normal subset as other GC B cells disappear in the $Prmt5^{F/F}$ Cγ1-cre mice. Alternatively, these may be cells failing positive selection and/or attempting differentiation. Defective B–T cell communication and/or signaling defect, as suggested by functional annotation of transcriptome alterations in Prmt5-deficient iGBs, may underlie this defect in vivo. The selection for Prmt5+ B cells in $Prmt5^{F/F}$ CD19-cre mice shows that the GC formation defect is B cell intrinsic. Competitive experiments would be necessary to formally rule out that the intrinsic defects in Prmt5-deficient B cell could cause secondary defects in other immune cells that might exacerbate the GC defect in with $Prmt5^{F/F}$ Cγ1-cre mice. In either case, our findings reveal that normal GC dynamics depends on Prmt5.

Mechanistically, we show that Prmt5 regulates the expression of a large number of genes in B cells. Despite we find Prmt5 mostly in the cytoplasmic of activated B cells, Prmt5-dependent nuclear sDMA is abundant, likely reflecting the largely repressive H3R8 and H4R3 methylation[24]. In addition, Prmt5 depletion reduces splicing fidelity, whereby weak donor splice sites are not recognized, producing RI and exon SE events that are likely to affect protein function. Prmt5 ensures spliceosome assembly by methylating SmD proteins[19,24], which are Prmt5 substrates in B cells (Supplementary Figs. 2A, 6J). While this function of Prmt5 has been observed in other systems[19,25,26], our results uncover the importance of splicing fidelity in B cell function and antibody responses. Splicing defects affect several histone modifiers, which likely contributes to the gene expression changes in Prmt5-null B cells. While this work was in revision, PRMT5 was found to methylate and thereby reduce the transcriptional repression activity of Bcl6 in human B cell lymphoma cell lines[55]. We did not detect changes in the Bcl6 signature in Prmt5-null iGBs (Supplementary Fig. 6C), but we cannot rule out that this may play a role in vivo during the GC reaction, which remains to be tested. Future work will dissect the indirect (via splicing or transcription factor regulation) and direct (via histone methylation at specific loci) contributions of Prmt5 to transcription regulation during B cell stage transitions. Unraveling Prmt5 function will require a comprehensive catalog of sDMA-modified proteins. Nonetheless, we establish the relevance of Prmt5-catalyzed sDMA in B cell biology and provide a framework to pursue the contribution of specific substrates to the p53-independent Prmt5 functions reported herein.

## Methods

**Animals.** Mice were housed at the specific pathogen-free facility of the IRCM. $Prmt5^{F/F}$ mice from EUCOMM[25] were backcrossed for >5 generations to C57BL6/ J. Cγ1-cre[32] and B1-8 knock-in[56] mice, kind gifts of Dr. K Rajewsky (MDC, Berlin), were obtained from Dr. Hua Gu at IRCM. CD19-cre mice[57] were a gift from Dr. Russell Jones (McGill University, Montreal). Mb1-cre mice[29] were a gift of Dr. Michael Reth (MPI, Freiburg). $Trp53^{−/−}$ mice[58] were obtained from Jackson labs (Bar Harbour, MN). $Cdkn1a^{−/−}$ mice[59] were a gift from Dr. Tarik Möröy (IRCM, Canada). $Aicda^{−/−}$ mice[60] were a gift from Dr. T. Honjo (Kyoto University, Japan). Aicda-GFP mice[61] were a gift from Dr. R Casellas (NCI Bethesda, MD). Lines combining alleles were generated by breeding at our animal house. Throughout the paper, the corresponding Cre driver $Prmt5^{+/+}$ mice were used as controls. Animal work was reviewed and approved by the animal protection committee at the IRCM (protocols 2013-18 and 2017-08) according to the guidelines of the Canadian Council on Animal Care.

**Immunization and infection.** Age- and sex-matched mice of 40–120 days of age were immunized either intraperitoneally with 50 μg NP$_{18}$-CGG (Biosearch Technologies) in Imject Alum adjuvant (Thermo Scientific) or intravenous with $10^9$ SRBC in 200 μL PBS (Innovative Research, IC100-0210). Mice were bled and/or sacrificed at various time post-immunization. Infections with *H. polygyrus bakeri* larvae were done by administering 200 L3 by gavage and mice were euthanized 14 days later.

**ELISA.** Sandwich ELISA for measuring pre-immune sera antibodies were done using plates coated with anti-isotype-specific antibodies (BD Pharmingen) to capture IgM, IgG1, IgG2b, or IgG3. Antigen-specific antibodies were captured from immunized mice sera in ELISA plates coated with NP$_{20}$-BSA (Biosearch Technologies) followed by the detection of IgG1. Detection was as described[62]. Briefly, antibodies were detected with the corresponding biotinylated anti-Ig (BD Pharmigen) followed by HRP-conjugated streptavidin (1:5000; Thermo Scientific) and developed using 2,2-azino-bis(3-ethylbenzothiazoline-6-sulfonic acid) substrate (Sigma).

**ELISPOT.** Purified splenocytes or BM cells were added at different dilutions to a 96-well 0.45-μm PVDF membrane (Millipore, cat#MSIPS4W10) previously coated overnight at 4 °C with 2 μg/mL NP$_{20}$BSA and blocked with complete RPMI cell culture media for 2 h at 37 °C. Plates with cells were incubated in a humid chamber 12 h at 37 °C, 5% CO$_2$, then washed 6× with PBS 0.01% Tween-20, followed by incubation with goat anti-mouse IgG1-HRP (A10551, Life Technologies, 1/2000) diluted in culture media for 2 h at RT. Plates were washed and AEC substrate (3′ amino-9-ethylcarbazole; BD Bioscience) was added to reveal the spots. Images were acquired in an Axiophot MZ12 microscope and scored spots were counted from the $2 \times 10^6$ cells dilution.

**Cell culture.** Naïve primary B cells were purified from splenocytes by depleting CD43+ cells using anti-CD43 microbeads (Miltenyi, cat. #130-049-801) and an autoMACS (Miltenyi). Primary B cells were cultured at 37 °C with 5% (vol vol$^{−1}$) CO$_2$ in RPMI 160 media (Wisent), supplemented with 10% fetal bovine serum (Wisent), 1% penicillin/streptomycin (Wisent), 0.1 mM 2-mercaptoethanol (bioshop), 10 mM HEPES, 1 mM sodium pyruvate. Resting B cells were stimulated with LPS (5 μg/mL, Sigma) + IL-4 (5 ng/mL, PeproTech). Induced germinal center B cells (iGBs) were generated using 40LB feeder cells[33]. Briefly, one day before B cell plating, 40LB cells were irradiated (120 Gy) and plated at $0.3 \times 10^6$ cells per well in 2 mL (6-well plate) or $0.13 \times 10^6$ cells per well (24-well plate) in 0.5 mL DMEM media supplemented with 10% fetal bovine serum (Wisent) and 1% penicillin/ streptomycin (Wisent). Purified naïve B cells were plated on 40LB feeders at $10^5$ cells per well in 4 mL of B cell media (6-well plate), or $2 \times 10^4$ cells per well in 1 mL of B cell media (24-well plate), supplemented with 1 ng/mL IL-4. At day 3 post-plating, the same volume of fresh B cell media was added to the wells, supplemented with 1 ng/mL IL-4 (PeproTech). On subsequent days, half of the volume per well was removed and replaced with fresh B cell media supplemented with cytokines. When re-plated, cells were harvested from the primary culture on day 4 and plated on fresh 40LB feeders in media supplemented with either 1 ng/mL IL-4 or 10 ng/mL IL-21. Re-plated cells were not fed and analyzed 3 days later. The Prmt5 inhibitor EPZ015666 (Cayman chemical, cat. # 17285) was aliquoted and resuspended for long-term storage at −80 °C at 50 mM in DMSO. Intermediate dilutions in DMSO (from 25 to 5 mM) were kept at −20 °C and frozen/thawed up to three times for individual experiments. DMSO was always diluted 1/1000 in final volume with cells.

**Flow cytometry.** Mononuclear cells from mouse spleen and MLN were obtained by mashing through a cell strainer with a syringe plunger. BM cells were obtained by opening and flushing femur and tibia bones from one leg with PBS, using a 1-mL syringe with 23G needle. BM and splenocytes suspensions were washed in PBS and incubated in 1 mL red blood cell lysis buffer (155 mM NH$_4$Cl, 10 mM KHCO$_3$, 0.1 mM EDTA) for 5 min at RT to complete lysis before filtering through 40-μm nylon cell strainer and resuspending in PBS 1% BSA. Single cell suspensions were stained with combinations of antibodies listed in Supplementary Data 5. To assess the proliferation in vivo, splenocytes were surface stained for GC markers and then $3 \times 10^6$ cells were treated with fixation/permeabilization solution (eBioscience, cat. #00-5123-43) for 1 h at 4 °C in the dark, washed twice in Perm buffer (eBioscience), followed by 1 h incubation with anti-Ki67-PECY7 (eBioscience) at 4 °C, and resuspended in PBS + 1% BSA. If necessary, anti-biotin staining was performed following Ki67 stain. To evaluate apoptosis in vivo, cells were treated with the FITC conjugated CaspGLOW reagents that detects activated pan-caspases (BioVision, K180) according to the manufacturer's instructions. Briefly, $10^6$ cells were treated with 2 μL FITC-VAD-FMK antibody for 1 h at 37 °C in 300 μL warm media, washed, surface stained and analyzed immediately. To assess apoptosis $0.3–0.5 \times 10^6$ cultured cells were stained with 3 μL Annexin V in 100 μL Binding buffer (cat. #51-66121E, BD Pharmigen) for 15 min at RT. Then, 400 μL of Binding buffer and 5 μL of propidium iodide (20 μg/mL) were added prior to flow cytometry acquisition. To assess γH2AX levels in vivo, $3 \times 10^6$ splenocytes were first surface stained for GC markers then washed twice in PBS 1% FCS 0.09% sodium azide and treated with fixation solution (BD, Cytofix #554655) for 30 min at 4 °C in the dark, then washed as previously. Pre-chilled Perm buffer III solution (BD#558050) was then added to the cells drop-wise under constant agitation. Cells were incubated for 30 min on ice in the dark and washed twice as previously before staining with anti-γH2AX (rabbit, 1/50) or anti-His control

(rabbit 1/50) in wash solution (final volume 100 μL) for 1 h at 4 °C in the dark and washed as previously. Cells were stained with the secondary antibody anti-rabbit Alexa633 (1/8000) in 100 μL wash solution for 1 h at 4 °C in the dark, washed and resuspended in PBS + 1% BSA. For intracellular IgG1 staining, cells were stained for GC markers then fixed/permeabilized according to the manufacturers protocol (eBioscience, cat. #00-5123-43) before staining for IgG1. To evaluate cell cycle profile, B cells were incubated with 10 μM BrdU for 1 h at 37 °C and fixed with ice cold 70% ethanol with constant agitation and incubated on ice for 30 min. Then 2 N HCl/Triton X-100 0.5% was added to the loosen cell pellet to denature the DNA. Cells were then washed, resuspended in 0.1 M $Na_2B_4O_7$, washed again and resuspended in PBS 0.5% Tween-20 1% BSA. $1 \times 10^6$ cells were then stained with anti-BrdU-FITC (1/50) for 30 min at RT in the dark and resuspend cells in PBS 5 μg/mL PI. Data were acquired using BD LSR Fortessa (BD biosciences) or BD Facscalibur (BD biosciences) and analyzed using FlowJo. For sorting, cells were stained as above and passed through a BD FACSARIA III (BD biosciences).

**Immunohistochemistry.** Section of 5-μm of paraffin-embedded tissues were deparaffinized in two changes of xylene for 5 min each and then rehydrated in distilled water using graded alcohols. Antigen retrieval was done by steaming the slides for 20 min then cooling for 20 min in either EDTA buffer (1 mM EDTA, 0.05% Tween 20, pH 8) for AID and PRMT5; or citrate buffer (10 mM acid citric, 0.05% Tween 20 pH 6.0) for PNA. Endogenous peroxidase was blocked with a 0.3% hydrogen peroxide solution for 10 min. When needed, endogenous biotin was blocked for 15 min with the blocking buffer provided with the Avidin/Biotin System (#SP2001, Vector Laboratories; Burlingame, CA). For protein block, we used 10% normal goat serum and 1% BSA for 60 min at RT or Carbo free buffer (#SP5040, Vector Laboratories). Sections were incubated with anti-AID (1:50, rat Mab mAID-2 eBioscience), anti-PRMT5 (1:250) overnight at 4 °C or biotinylated PNA (1:100) for 60 min at RT. Biotin-conjugated secondary antibodies were goat anti-rabbit IgG (1:200, Vector laboratories) to detect anti-PRMT5; goat anti-rat IgG (1:200, Vector laboratories) to detect anti-AID. Biotinylated reagents were detected with Vectastain ABC kit (PK-6100, Vector laboratories). Peroxidase activity was developed using ImmPACT NovaRED HRP substrate (Vector laboratories). The sections were counterstained with hematoxylin (Sigma cat. #MHS32-1L) for 1 min prior to dehydrating and mounting. The antibodies are listed in Supplementary Data 5.

**Immunofluorescence.** Tissues were frozen in OCT (VWR #95057-838). Sections of 5-μm were fixed in PFA 4% for 10 min at RT, washed three times in PBS at RT, followed by an incubation in pre-chilled acetone for 10 min at −20 °C. Sections were permeabilized in 0.5% Triton X-100 in PBS for 10 min at RT. Sections were then blocked in PBS, 5% goat serum, 1% BSA, 0.3% Triton X-100 for 1 h at RT. Incubations with primary antibody were performed in blocking solution overnight at 4 °C in a humid chamber in the dark and, when needed, secondary antibody was added for 1 h at RT. Primary B cells were washed 1× with PBS and then plated on coverslips coated with 0.1 mg/mL poly-L-lysine (Sigma). Cells were centrifuged 5 min at $400 \times g$, then allowed to adhere at 37 °C for 20 min, before fixation with 3.7% formaldehyde (Sigma) for 10 min at RT. After three washes with PBS, coverslips were blocked for 1 h with blocking solution (5% goat serum, 1% BSA, 0.5% Triton X-100 in PBS). Cells were then incubated overnight at 4 °C with anti-PRMT5 (1:100) or anti-SYM11 (1:100), diluted in blocking solution. After 3 × 5 min washes, with PBS + 0.1% Triton X-100 (PBS-T), cells were incubated for 1 h at RT with anti-rabbit IgG Alexa-546 (1:500) diluted in blocking solution. After 3 × 5 min washes with PBS-T, cells were incubated with 300 nM DAPI (ThermoFisher) in PBS for 5 min at RT. Finally, coverslips were washed with PBS followed by ddH₂O before mounting onto slides using Lerner Aqua-Mount (ThermoFisher). The antibodies are listed in Supplementary Data 5.

**Microscopy and image analysis.** Confocal microscopy was acquired using a Zeizz LSM519 Axiovert 100 M with 20X LD-Achroplan or 40X Plan-Neofluor (with oil immersion) objectives. Prmt5 and Sym11 signal in B cells was scored in Volocity (Perkin Elmer), quantifying signal within a dilated mask generated using DAPI. For each experiment, multiple fields were analyzed, excluding cells with saturated signal, abnormal DNA structures or mitotic figures. Fluorescent microscopy images were acquired in a Leica DM6000 microscope equipped with a Hamamatsu Orca-ER C4732 camera. Images were acquired with a Leica 40X HCX PL Fluotar objective or 63X HCX PL APO (with oil immersion). Quantification of γH2AX foci was performed on fluorescent microscopy images, using a series of image processing operation to separate B-cell DNA damage from the larger feeder cells. Analysis scripts were programmed using Matlab 2016a (Mathworks Inc.). A total of 42–48 images per genotype per day were automatically analyzed for >900 cells per sample. Cell nuclei were found using the DAPI channel of images, based on an Otsu intensity threshold. Binary images were cleaned of small objects, holes were filled, and edges were smoothed using a morphological closing operation with a 5-pixel-radius circular kernel. Multinuclear cells were removed from this nuclear mask by computing the convex Hull of all objects, and objects where the ratio of their area over their convex Hull area was lower than 0.8 were discarded. Finally, big nuclei, belonging to feeder cells were discarded by only keeping objects of area smaller than 2% of the image size. DNA damage foci were assumed to be fluorescent puncta, most of them of sub-diffraction-limit size. Intensity contrast was enhanced by saturating the bottom 1% and the top 1% of all pixels. Fluorescent puncta were detected using a linear band-pass filter that preserved objects of a chosen size and suppressed noise and large structures. After the filter, we chose elements bigger than noise up to twice the diffraction limit. Only foci detected within B-cell nuclei were considered for statistical purposes. In order to consider only foci brighter than in control IF, we quantified the average foci intensity of non-treated cells, and only foci of higher-than-control intensity were considered to compute the number of DNA damage foci per cell. Prmt5 quantification in immunized Prmt5F/F CD19-cre B cells was done on fluorescence microscopy images of single follicles containing 1 or 2 GC. A first approximation for cells segmentation was obtained using a Laplacian of the Gaussian filter followed by a median filter and a watershed transformation. A typical cell size was manually established for all images. Segmented objects of area larger than five times the typical cell were discarded from analysis. Objects of area smaller than half a cell were fused with the adjacent cell of largest eccentricity. Typically, 2000 cells were detected and processed in each image. To assign nuclear and cytoplasmic pixels to each cell, the typical radius of a circular cell of the manually set area was calculated. For each cell, a band was established around its edge of thickness radius/3 and assigned to the cytoplasm all pixels on the band. The median fluorescence in the cytoplasm of each one of the channels was computed for every cell. The GCs and surrounding B cell follicle were manually delineated using the IgD and GL7 IF images to create two masks, and cells were assigned to either the GC or the follicle. Cell with pixels spilt between those two masks were assigned to the mask of more pixels. The fluorescence signal for Prmt5 of all cells was normalized to the cell with maximal signal in the corresponding image, after eliminating outliers using a histogram of the distribution (example in Supplementary Fig. 1D). The normalized values for all cells in a follicle and its associated GC were averaged to obtain the normalized mean intensity per cell in each compartment. The normalized mean of all follicles from a single mouse were averaged, and the same was done for the GC. These values were plotted and used to calculate ratios in Fig. 2b.

**Western blots.** Protein extracts were performed by lysing cells in NP-40 lysis buffer (1% NP-40, 20 mM Tris pH 8, 137 mM NaCl, 10% glycerol, 2 mM EDTA), containing protease and phosphatase inhibitor (Thermo Scientific). Protein extracts were separated by SDS-PAGE and transferred to nitrocellulose membranes (BIO-RAD). In some experiments, equal protein loading was controlled by staining the membrane after transfer using REVERT total protein stain solution (LI-COR). Membranes were blocked in TBS 5% milk and probed with primary antibodies overnight, washed 4 × 5 min in TBS + 0.1%Tween before incubating with secondary antibodies conjugated to AlexaFluor680 or IRDye800 for 1 h, washed and read on Odyssey CLx imaging system (LI-COR). Proteins were quantified using ImageStudiolite software. When Revert staining (LI-COR) was used, Revert signal from a whole protein lane was considered for normalization. The antibodies are listed in Supplementary Data 5. The original uncropped images of all western blots are shown in Supplementary Fig. 7.

**RT-qPCR.** RNA isolated using TRIzol (Life Technologies) was reverse transcribed with ProtoScriptTM M-MuLV Taq RT-PCR kit (New englandBiolabs). Quantitative PCR using SYBR select master mix (Applied Biosystems) was performed and analyzed in a ViiATM 7 machine and software (Life technologies). Oligos used for PCR are listed in Supplementary Data 6.

**RNA-seq.** Cγ1-cre and Prmt5F/F Cγ1-cre splenic B cells plated on 40LB cells were separated from the feeder cells by elutriation at day 4 after plating. Total RNA was extracted from four independent biological samples per genotype, using RNeasy plus Mini kit (Quiagen), quality controlled using RNA Pico chip on Bioanalyzer (Agilent) and processed individually for RNA-seq. Samples were depleted of rRNA using Ribo-zero Gold rRNA removal kit H/M/R (Illumina Hu Mm Rt). Libraries were constructed using the KAPA stranded RNA-Seq kit (Roche), which included fragmentation, cDNA, dsDNA, ER, ATL, Ligation and PCR enrichment, and purified by KAPA magnetic beads (Roche). The mean fragment size obtained was 308 bp. QC validation was performed on HSdna chip on Bioanalyzer and Q-PCR titration with NEBnext Library quant Kit for Illumina (NEB). Sequencing PE125 on HiSeq2500 with v4 chemistry (Illumina) at the Genome Center (Quebec).

**RNA-seq analysis.** Sequencing reads were trimmed using Trimmomatic (v0.32)[63], removing adaptor and other Illumina-specific sequences as well as the first four bases from the start of each read, and low-quality bases at the end of each read, using a 4 bp sliding window to trim where average window quality fell below 30 (phred33 < 30). An additional 3 bp were clipped from the start and end of a read if found of low quality. Trimmed reads <30 bp were discarded. The resulting clean set of reads were then aligned to the reference mouse genome build mm10 using STAR (v2.3.0e)[64] with default parameters. Reads mapping to more than 10 locations in the genome (MAPQ < 1) were discarded. Gene expression levels were estimated by quantifying uniquely mapped reads to exonic regions (the maximal genomic locus of each gene and its known isoforms) using featureCounts (v1.4.4)[65] and the Ensembl gene annotation set. Normalization (mean of ratios) and variance-stabilized transformation of the data were performed using DESeq2[66]. Differential

expression analysis was performed with DESeq2 using the Wald test. Statistically significant (adjusted $p$ value < 0.05, Benjamini–Hochberg procedure) genes with large expression changes (absolute fold change ≥1.5) that are expressed above a threshold (average normalized expression across samples >50) were considered as differentially expressed genes. Multiple control metrics were obtained using FASTQC (v0.11.2), samtools (v0.1.19), BEDtools (v2.17.0), and custom scripts. For visualization, normalized Bigwig tracks were generated using BEDtools and UCSC tools. Integrative Genomic Viewer was used for data visualization. For splicing analysis, we used rRMATS v3.2.5[67] with default parameters to identify alternative splicing (AS) events. Since rMATS requires reads of equal lengths, we performed the analysis on untrimmed reads, using the Mus musculus GRCm38 release 87 Ensembl gtf as exon annotation file. We used both rMATS outputs for analysis: AS events identified using reads mapping to splice junctions and reads on target (JC + ROT), and AS events identified using reads mapped to splice junctions only (JC). Each of the five types of AS events (skipped exon, SE; mutually exclusive exon, MXE; retained intron, RI; alternative 5' splice site, A5SS; and alternative 3' splice site, A3SS) was analyzed separately. A non-redundant list of genes displaying significant AS events was generated by selecting the most statistically significant AS event per gene with more than 10% inclusion level difference. Benjamini–Hochberg correction was applied to this non-redundant set of events. The final list of AS affected genes (Supplementary Data 3) contains events with FDR < 0.05 in genes that pass a minimum expression threshold (normalized mean expression >50).

**Functional annotation**. GO terms enrichment was performed using either DAVID (https://david.ncifcrf.gov/)[68] or Gorilla (http://cbl-gorilla.cs.technion.ac.il/)[69] by providing either an unranked gene set and a background list, or a ranked list, as follows. The lists of genes differentially expressed by ≥2-fold in either direction with $p$-adj < 0.05 were analyzed by DAVID, using the list of all expressed genes (basemean > 0) as background, to identify enriched GO biological process terms. A non-redundant list of aberrantly spliced genes reaching statistical significance was similarly analyzed at Gorilla. GO or KEGG with FDR < 25% and $p$ < 0.05 in adjusted $p$-value were considered significant. Gorilla permitted the analysis of ranked lists. Two list were compiled from differential expression analysis, ranking genes according to log2fold-change, from the most changed to the last unchanged gene in each direction (i.e. the list with upregulated genes excluded downregulated genes and vice versa). Each list was uploaded to Gorilla. In all methods, GO terms with $p$ < 0.05 and FDR $q$-value < 0.05 were considered significant. A list containing gene expression data for each sample for genes with average basemean >20 was analyzed by GSEA[70] using the interface v3.0 provided by the Broad Institute (http://software.broadinstitute.org/gsea/index.jsp), using the default KEGG list of gene sets at MsigDB[71]. The analysis was run with 1000 permutations, excluding gene sets containing more than 1000 and less than 15 genes, ranking genes by Signal2Noise and using weighted enrichment statistic. Specific gene expression signatures for selected candidate pathways to analyze by GSEA were obtained from the literature, as reference in results, and analyzed using the same input data and parameters.

**Reporting Summary**. Further information on experimental design is available in the Nature Research Reporting Summary linked to this article.

## Data availability

RNA-seq data generated in this work and used for Fig. 6a–h, Supplementary Figs. 5b–e, 6c, and 6e are available at NCBI under accession number GSE120309. Accession numbers of RNA-seq or microarray datasets obtained from public repositories are provided in the corresponding figures or figure legends. All other relevant data are included. Ad hoc Matlab code produced for the quantification of IF signals in Fig. 2b, Supplementary Figs. 1d, and 5f described above is available upon request.

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

## Acknowledgements
We thank Dr. A. Veillette and Dr. Hua Gu for critical reading. We thank Dr. T. Moroy, D. Scott, J. Fraszczak, M. Rashkovan, C. Vadnais, M. Lapointe for technical help, reagents, and advice; M. Cawthorn, E.-L. Thivierge, M. Laprise, M.-C. Lavallée, and S. Demontigny for animal technical help; E. Massicote and J. Lord for help with flow cytometry; S. Terouz with immunohistochemistry, D. Filion with microscopy imaging, O. Neyret, M. Rondeau, A. Dumont with RNA-seq. This work was supported by operating grants from The Cancer Research Society, Inc. (#18075 and #20194 to J.M.D.N.) and Canadian institutes of health research (MOP 125991 to J.M.D.N., MOP-130579 to I.L.K.). We acknowledge the doctoral fellowships from Fonds de recherche du Québec—Santé (FRQ-S) (to A.P.M. and S.P.M.) and Cole foundation (to S.P.M. and L.C.L.). I.L.K. is a Canada Research Chair in Barrier Immunity. J.M.D.N. was supported by a Canada Research Chair in Genetic Diversity and is a Senior FRQ-S research scholar.

## Author contributions
Conceptualization: L.C.L., J.M.D.N.; methodology: L.C.L., A.Z., S.C.; investigation: L.C.L., A.Z., A.M., S.H., A.M.P., S.P.M., A.S., T.B.; reagents: D.K., S.R.; data curation: S.H., C.K., I.K., S.R., L.L., J.M.D.N.; writing—original draft: L.C.L., J.M.D.N. All authors discussed and commented on the manuscript. Funding acquisition: J.M.D.N.

## Additional information

**Competing interests:** The authors declare no competing interests.

