## [Peer Review File · Nature Communications]

Reviewers' comments:

Reviewer #1 (B cell subsets, methylation)(Remarks to the Author):

Litzler et al describe the multiple roles for the arginine methyl transferase PRMT5 in B cell development and differentiation. They show expression of the enzyme in the various stages of development in adults, increased expression on activation, the consequences of gene deletion on development, in vivo and in vitro differentiation and proliferation, gene expression and the partial compensation through removal of p53, the tumour suppressor. This really is a most comprehensive analysis with multiple aspects of the phenotype described and investigated at considerable detail both through in vitro experiments and the use of multiple knockout models. Perhaps not surprisingly for an enzyme with such a large and varied number of substrates in a large number of cells, the knockout is very pleiotropic. Despite this, Litzler et al have revealed and partially characterised a really interesting set of phenomena, which will keep interested B cell biologists, PRMT specialists and those interested in the integration of translational and post-translational regulation of cell function. This really is an excellent study and in my opinion very well suited to Nature Communications, more or less as is. The experiments have been well conducted and the data well interpreted, it is clearly and precisely written, the conclusions are based on the data and the discussion is intelligent and well rounded.

There are, however, some minor issues that the authors should address for the sake of completeness and to stop minor errors detracting from what is a seminal piece of work. In order of appearance:

1. Please define type III methyltransferases in the Introduction
2. The reference to Supp Fig 5D on page 15, should be to 6D.
3. P18 discussion, first paragraph. The deletion of PRMT5 after B cell activation only not "prevents apoptosis" relative to the pre-activation deletion of PRMT5, not in absolute terms.
4. P19 "Our data indicates ..." change to "Our data indicate ..."
5. P19 "... with PRMT5-null be B cells ..." change to "null B cells"
6. Figure 1 C. How mature B cells differ from Fo B cells and how these numbers differ?
7. Figure 2B – is there a better example of GL7 staining in immunofluorescence? The one shown is a long way from the FACS calculation of percent positive.
8. I can't determine what is different between Fig 2F and 2G.
9. In Fig 4B (and throughout) please specify the time point used for these data
10. Fig 4G shows GC frequency in KO to be 1/6th of WT, but in Fig 2F these frequencies were more or less the same. Is there something different?
11. Fig 5G. The CXCR4- population is very interesting. One thing to be careful of in using the percent representation is that in the KO mice, the GC number has collapsed around the time that these cells reach prominence. Is it that a number of these cells are generated equally in both and then maintained in both strains, but with the disappearance of the conventional GC cells in the KO, their percent goes way up?
12. Fig 6I – is the frequency of gH2AX+ cells always increased in the KO GC? Even early?
13. Fig 8H doesn't show the S phase deficit that was apparent in Fig 3H. What is different?
14. Fig 9F – can you detail the time after immunisation and antigen for FACS panels?
15. Fig 9G change LPA to LPS
16. Fig 9H change "as in H)" to "as in G)"
17. Fig 9J change "as in G)" to "as in I)"

Reviewer #2 (Plasma cell differentiation)(Remarks to the Author):

Litzler et al. provide a thorough experimental plan to investigate the role of Prmt5 in B cell

development and differentiation. As known, Prmt5 is the major enzyme in symmetric arginine dimethylation in cells and affects many pathways. They find that Prmt5 is important and all key steps in B cell development and in the process of differentiation into plasma cells, where the key result is reduced numbers of cells or increased apoptosis. At early stages of development, they find a p53 dependent defect, while at later stages, they find p53 independent defects. They do a terrific job at trying to subset the stages using a variety of Cre mice that highlight specific steps.

Major comments:

1. The authors claim that Prmt5 B cell defects are intrinsic. This is based on the use of Cre mice and ex vivo experiments. Although they are likely correct, for this claim to be made, the authors will have to at least do an adoptive transfer (OK) or a mixed bone marrow chimera (best) experiment in which the deficient and sufficient cells are present in the same animal.
2. Numbers of animals used. The numbers of animals used in the experiments presented is highly variable and in many experiments is too few. Phenotypic experiments should have at least 4-5 mice / group to normally fit a standard power analysis. Certainly those with 2 mice each would not allow statistics to be performed.
3. The number of independent samples in each group for the RNA seq was not described (or I missed it), hopefully this was at least 2.

Minor comments

The interpretation of the GSEA plots in fig 9c does not appear to be accurate. For some there is a clear skewing and representation to either the deleted allele or control as the authors state. But for others, such as mTORC1, Plasma/memory and perhaps the EZH2 targets it appears as though the Prmt5 deletion dysregulates these genes through different mechanisms as there are increases in genes on both sides of the plots.

Reviewer #1

1. Please define type III methyltransferases in the Introduction.

This is now included.

2. The reference to Supp Fig 5D on page 15, should be to 6D.

Thank you for pointing this out. It has been changed.

3. P18 discussion, first paragraph. The deletion of PRMT5 after B cell activation only not “prevents apoptosis” relative to the pre-activation deletion of PRMT5, not in absolute terms.

We agree. We have rephrased to accurately described the effects.

4. P19 “Our data indicates ...” change to “Our data indicate ...”

5. P19 “... with PRMT5-null be B cells ...” change to “null B cells”

Both changes were done.

6. Figure 1 C. How mature B cells differ from Fo B cells and how these numbers differ?

Mature B cells includes follicular and MZ B cells. The sum of follicular + MZ B cells is not exactly equal to the number of mature B cells because they were calculated from two different flow cytometry stainings, each gated on B220+ cells. Both gating strategies are shown in Supplementary Figure 1B. We have modified panel 1C to separate the populations according to each staining.

7. Figure 2B – is there a better example of GL7 staining in immunofluorescence? The one shown is a long way from the FACS calculation of percent positive.

We have changed the pictures as recommended. Note that the example was not selected to coincide with the flow cytometry data but to illustrate that the residual Prmt5+ follicular B cells in *Prmt5^{F/F}* CD19-cre mice are highly selected to form GC in these mice. Answering to a concern of Reviewer 2, we have done additional IF in immunized mice with higher quality images at higher magnification. We used examples from these new staining in 2B, and added the new supplementary Fig 1D that are consistent with the FACS in 2A for *Prmt5^{F/F}* CD19-cre and control mice.

8. I can't determine what is different between Fig 2F and 2G.

Thank you for pointing this out, it was indeed confusing. The original 2G and 2F were different analyses of apoptosis using different gating on the same data, with F included to stratify non-GC B cells according to GL7 levels.

We have simplified the figure by rearranging and merging the two panels into new panel 2F. The new panel shows an example of the gating strategy on top, with representative histograms of activated pan-caspase staining for each cell subset in F/F and Ctrl cells. The plots at the bottom compile data from multiple mice. Conclusions remain the same.

9. In Fig 4B (and throughout) please specify the time point used for these data

Time points have been added to the legends of Fig. 4 and wherever necessary.

10. Fig 4G shows GC frequency in KO to be 1/6th of WT, but in Fig 2F these frequencies were more or less the same. Is there something different?

They are different. Both panels analyze *H. polygyrus* infections at day 14 but in mice of different genotypes. Fig. 4G shows data from *Prmt5^{F/F} Cg1-cre* mice, in which *Prmt5* is ablated in GC and causes a large reduction in GC B cell proportion and number, as reflected in the 4G panel. Fig. 2F showed data from *Prmt5^{F/F} CD19-cre* mice, in which GC B cells are reduced in number but not proportion (because the *Prmt5⁺* escapees in *Prmt5^{F/F} CD19-cre* are strongly selected to form GC, which together with the general loss of splenic B cells upon infection in these mice explains the normal proportions). This panel was modified as explained above, but we included the data below from multiple *Prmt5^{F/F} CD19-cre* mice for the record.

Legend: GC B cell number in MLN of and CD19-cre *Prmt5^{F/F}* and CD19-cre mice infected with *H. polygyrus*, day 14 post infections. Values for individual mice and means are plotted.

11. Fig 5G. The CXCR4- population is very interesting. One thing to be careful of in using the percent representation is that in the KO mice, the GC number has collapsed around the time that these cells reach prominence. Is it that a number of these cells are generated equally in both and then maintained in both strains, but with the disappearance of the conventional GC cells in the KO, their percent goes way up?

We agree with the Reviewer on this point. We have added this possibility in the revised Discussion.

12. Fig 6I – is the frequency of gH2AX+ cells always increased in the KO GC? Even early?

We had only checked this at day 10. While now measured gH2AX at day 5 and did not detect a difference in DNA damage between control and *Prmt5^{F/F} Cg1-cre*, in line with the relatively normal number of GC at this time point. We speculate that DNA damage accumulates with time.

13. Fig 8H doesn't show the S phase deficit that was apparent in Fig 3H. What is different?

Fig. 8H analyzed Cg1-cre *Prmt5^{F/F} Trp53^{-/-}* mice. The S phase deficit trend is apparent but does not reach significance because of the low n (=3). We are confident that increasing the n will make the difference statistically significant but, unfortunately, we discontinued this line a long time ago. Since the trend is visible and removing p53 does not rescue proliferation or any GC phenotype, we do not believe that the long delay required to remake this line is justified for this relatively small point.

14. Fig 9F – can you detail the time after immunisation and antigen for FACS panels?

We have added a label to indicate that the panels shown are from day 6 after SRBC immunization.

15. Fig 9G change LPA to LPS

16. Fig 9H change “as in H)” to “as in G)”

17. Fig 9J change “as in G)” to “as in I)”

Thank you. All were corrected.

Reviewer #2

Major comments:

1. The authors claim that *Prmt5* B cell defects are intrinsic. This is based on the use of Cre mice and *ex vivo* experiments. Although they are likely correct, for this claim to be made, the authors will have to at least do an adoptive transfer (OK) or a mixed bone marrow chimera (best) experiment in which the deficient and sufficient cells are present in the same animal.

We accept the criticism. We only claimed B cell intrinsic effects for *Prmt5* in GC formation and antibody responses. Our experience with adoptive transfer of B cells into μ MT mice has been disappointing. At least in the conditions we tested (2×10^7 splenic B cells transferred + IP immunization and analysis 15 days later), there were very few if any GC-like cells by flow cytometry, the surface marker profile was quite different from the real thing and there were no discernible GC structures in the spleen, so the results were uninformative. The appropriate experiments would be as suggested, BM chimeras to compare *Prmt5* sufficient and deficient GC B cells, using *Prmt5^{F/F} Cg1-cre* or *Prmt5^{F/F} CD19-cre* mice. These experiments would take at least 3 months all going well; but obtaining the necessary mice would have taken us an additional 3 months. We wanted to avoid such a delay given the overall positive comments from both Reviewers but also out of timing concerns due to potentially overlapping papers that were under consideration while we did these revisions (see Discussion). For these reasons, we have 1) moderated our conclusion from the *Prmt5^{F/F} Cg1-cre* mice and, 2) provided experimental support for a B cell intrinsic defect in GC formation in *Prmt5^{F/F} CD19-cre* mice.

1) The experiments from which we originally concluded B cell intrinsic effects for GC formation and antibody responses (Fig. 4) were performed with *Prmt5^{F/F} Cg1-cre* mice. We note that the Cg1-cre driver is very specific, as the Cg1 region is not transcribed outside activated B cells in immune cells (Ref. Cassola *et al*, 2006). A small proportion of mice can delete in the germ cells, but we always check for excision in tail tissue to avoid using mice derived from such events, although we have hardly ever observed this. Because of this tight tissue specificity, people very rarely perform BM chimeras to confirm B cell intrinsic effects when using the Cg1-cre. Furthermore, we provide all the controls to show that T cell populations are not affected. Moreover, the paper contains many experiments *ex vivo* that demonstrate B cell intrinsic roles of *Prmt5* in apoptosis, cell proliferation, transcription and splicing; which could explain many of the defects *in vivo*. Hence, as the Reviewer acknowledges, the defects observed in *Prmt5^{F/F} Cg1-cre* mice, are very likely to be B cell intrinsic. Nonetheless, because

we do not have the BM experiment, we have revised the conclusion from figure 4 to state that the effects are likely to be B cell intrinsic.

2) We did analyze an unconventional but valid competitive system, which was afforded by the efficient but still partial Prmt5 ablation in the population of resting B cells of *Prmt5^{F/F}* CD19-cre mice. Indeed, measuring Prmt5 mRNA and protein levels shows that a majority, but not all, of resting follicular B cells are Prmt5-negative in *Prmt5^{F/F}* CD19-cre mice. Yet, after immunization most if not all GC B cells in *Prmt5^{F/F}* CD19-cre mice are Prmt5+ (see **new Fig. 2B** and **new supplementary Fig. 1D**).

To provide statistical support to this observation, as requested by the Editor, we have immunized additional *Prmt5^{F/F}* CD19-cre mice and taken higher quality microscopy pictures to quantify Prmt5 by IF in follicular and GC B cells using an ad hoc script. The data (**new Fig 2B**) demonstrates that the minority of Prmt5+ resting B cells remaining in the follicles of *Prmt5^{F/F}* CD19-cre mice outcompete the majority of Prmt5- B cells to form GC, implying a B cell intrinsic effect.

The quantification strategy is described in Methods. Briefly, we calculated the normalized intensity value, which is less arbitrary than calling positive and negative cells. The normalized intensity can also be compared between different pictures. Thus, the averages of the normalized means from 5-8 individual follicles for 4 *Prmt5^{F/F}* CD19-cre and 4 CD19-cre mice are shown in **new Fig. 2B**. We present **a figure for the Reviewer** (below) with the mean + SD normalized intensity value for each mouse, which reflects well the expected variable level of excision in follicular B cells between *Prmt5^{F/F}* CD19-cre mice (compare with the values for CD19-cre controls), and clearly shows the selection for Prmt5+ B cells in the GC. The latter is better shown by the Prmt5 signal ratio between follicular and GC B cells in **new Fig. 2B**. An example of the distribution of normalized Prmt5 intensity in each individual cell scored in representative follicles and GC shown in **supplementary Fig. 1D**.

Legend: The box plots indicate the 10-90 percentiles and whiskers indicate minimum to maximum values of mean normalized Prmt5 IF intensity per cell for 5-8 B cell follicles (FO) and their associated GC B cells in individual CD19-cre (ctrl) and *Prmt5^{F/F}* CD19-cre (ko) mice at day 14 post immunization.

Thus, the new competitive experiments with *Prmt5^{F/F}* CD19-cre mice and the tissue specificity of the *Prmt5^{F/F}* Cg1-cre, together with the many effects shown *ex vivo* in B cells from these mice, strongly suggest that Prmt5 has B cell intrinsic functions required for GC formation and antibody response. Since we have nonetheless tone down our conclusions regarding B cell intrinsic effects, we would hope avoiding several months of additional BM experiments to definitively prove this very likely point if the Reviewer is satisfied with this answer.

2. Numbers of animals used. The numbers of animals used in the experiments presented is highly variable and in many experiments is too few. Phenotypic experiments should have at least 4-5 mice / group to normally fit a standard power analysis. Certainly those with 2 mice each would not allow

statistics to be performed.

We have repeated several experiments to bring the n up to at least 4 mice in most panels. New data was added to **figures 2A, 2B, 3A, 3E, 3F, 5A, 5F, 5G, 5K, 8C, 9E, 9F, 9H, 9J and supplementary figure 6G**. In all cases the conclusions remain the same after adding more animals, which were now tested statistically in each case.

Note that panel Fig. 9F was modified. The original panel showed that the proportion of IgG1⁺ plasma cells in the spleen of mice immunized with SRBC and mice infected with *H. polygyrus*. We have removed the latter to make room for data on the SRBC-immunized mice. This was because a) we could more easily increase the n; b) in the immunized mice we had also measured the proportion of IgM⁺ plasma cells, which we also repeated. The results show that *Prmt5^{FF}* Cg1-cre mice show not only reduced % of IgG1+, but also concomitantly increased % of IgM⁺ splenic plasma cells compared to controls. This result more strongly supports our proposal that they differentiate prematurely, before having the chance to switch.

We could not increase the n for figure panels 5J and 8G-H, depicting data from *Prmt5^{FF} Aicda^{-/-}* Cg1-cre and *Prmt5^{FF} Trp53^{-/-}* Cg1-cre mice, because we had discontinued these two lines and remaking them would take >6 months. Since, the data (with 3 mice each) is very clear and either not central to the story (*Aicda^{-/-}*), or the same conclusion was obtained from additional systems (*Trp53^{-/-}*), we believe that the investment would not be justified.

3. The number of independent samples in each group for the RNA seq was not described (or I missed it), hopefully this was at least 2.

We used 4 independent samples of each. Some of the heatmaps showed the 4 replicates (e.g. Sup figs 5C and 6C) but it is true that the number of samples was never mentioned. We added this information to the corresponding section of methods.

Minor comments

The interpretation of the GSEA plots in fig 9c does not appear to be accurate. For some there is a clear skewing and representation to either the deleted allele or control as the authors state. But for others, such as mTORC1, Plasma/memory and perhaps the EZH2 targets it appears as though the *Prmt5* deletion dysregulates these genes through different mechanisms as there are increases in genes on both sides of the plots.

We see the point of the reviewer, and we have toned down some of the claims related to GSEA analysis, for which we have rewritten the corresponding paragraphs in the last section of results and in the discussion. For large gene sets, like EZH2 and Plasma/memory, the bimodal enrichment at either side of the curve most likely reflects subsets of genes with distinct regulatory mechanisms, as the reviewer suggests. While the net result is that the gene sets were significantly enriched among the genes induced in *Prmt5*-null cells, we changed the claim to state that the enrichments were modest, which is not unexpected considering that RNA-seq was from day 4 iGBs when increased plasma cell differentiation is not yet apparent (**modified supplemental figure 5A**). In the case of mTORC1, on the other hand, we find the skewing of the curve quite clear and the enrichment statistically significant ($p < 0.001$, and normalized enrichment -2.1), and thus stand by our original interpretation. In any case, we do not base our overall conclusion about the propensity for plasma cell differentiation in any single GSEA result, but on the accumulated analysis of multiple signatures, the expression profile of *Prmt5* in the LZ, as well as phenotypic and experimental data that are consistent with this conclusion.

REVIEWERS' COMMENTS:

Reviewer #1 (Remarks to the Author):

I am happy with the response of the authors to my concerns and I think the manuscript is now acceptable. I think one has to balance certainty with time and cost and I am happy with the way the experiments have been done with a B cell intrinsic Cre. As mentioned by referee 2, I agree that there could be complex, second order interactions occurring, where deletion of PRMT5 in B cells causes a change that in turn causes a defect in Tfh (for example), that is then read out as a germinal center deficit. To a degree, this is still B cell intrinsic from the point of PRMT5 requirement. It may need to be revisited in future, but for the initial description of this important strain, I am comfortable with this provided the discussion of the result reflects this possibility of other cells being involved.

Response to Reviewers

Reviewer #1 (Remarks to the Author):

I am happy with the response of the authors to my concerns and I think the manuscript is now acceptable. I think one has to balance certainty with time and cost and I am happy with the way the experiments have been done with a B cell intrinsic Cre. As mentioned by referee 2, I agree that there could be complex, second order interactions occurring, where deletion of PRMT5 in B cells causes a change that in turn causes a defect in Tfh (for example), that is then read out as a germinal center deficit. To a degree, this is still B cell intrinsic from the point of PRMT5 requirement. It may need to be revisited in future, but for the initial description of this important strain, I am comfortable with this provided the discussion of the result reflects this possibility of other cells being involved.

We agree with the comment and have added a sentence to the discussion: "Competitive experiments would be necessary to formally rule out that the intrinsic defects in Prmt5-deficient B cell could cause secondary defects in other immune cells that might exacerbate the GC defect in with Prmt5^{F/F} Cg1-cre mice".

Reviewer #2 did not send comments.

We thank both Reviewers for their suggestions to improve our manuscript.